# AuToMATo: An Out-Of-The-Box Persistence-Based Clustering Algorithm

## Abstract

We present AuToMATo, a novel clustering algorithm based on persistent homology. While AuToMATo is not parameter-free per se, we provide default choices for its parameters that make it into an out-of-the-box clustering algorithm that performs well across the board. AuToMATo combines the existing ToMATo clustering algorithm with a bootstrapping procedure in order to separate significant peaks of an estimated density function from non-significant ones. We perform a thorough comparison of AuToMATo (with its parameters fixed to their defaults) against many other state-of-the-art clustering algorithms. We find not only that AuToMATo compares favorably against parameter-free clustering algorithms, but in many instances also significantly outperforms even the best selection of parameters for other algorithms. AuToMATo is motivated by applications in topological data analysis, in particular the Mapper algorithm, where it is desirable to work with a clustering algorithm that does not need tuning of its parameters. Indeed, we provide evidence that AuToMATo performs well when used with Mapper. Finally, we provide an open-source implementation of AuToMATo in Python that is fully compatible with the standard *scikit-learn* architecture.

## 1 Introduction

Clustering techniques play a central role in understanding and interpreting data in a variety of fields. The idea is to divide a heterogeneous group of objects into groups based on a notion of similarity. This similarity is often measured with a distance or a metric on a data set. There exist many different clustering techniques (Anderberg, 1973; Duda et al., 2000), including hierarchical, centroid-based and density-based techniques, as well as techniques arising from probabilistic generative models. Each of these methods is proficient at finding clusters of a particular nature. Many of the most commonly used clustering algorithms require a selection of a parameters, a process which poses a considerable challenge when applying clustering to real-world problems.

In this work, we present and implement AuToMATo (*Automated Topological Mode Analysis Tool*), a novel clustering algorithm based on the topological clustering algorithm ToMATo (Chazal et al., 2013). The latter summarizes the prominences of peaks of a density function in a so-called persistence diagram. The user then selects a prominence threshold $\tau$ and retains all peaks whose prominence is above this threshold, which results in the final clustering. A simple heuristic to select $\tau$ is to sort the peaks by decreasing prominence, and to look for the largest gap between two consecutive prominence values (Chazal et al., 2013). While yielding reasonable results in general, this procedure is not very robust to small changes in the prominence values.

A more robust and sophisticated method is to perform a bottleneck bootstrap on the persistence diagram produced by ToMATo, which is precisely what AuToMATo does. That is, given a persistence diagram obtained by running ToMATo on a point cloud, AuToMATo produces a confidence region for that diagram with respect to the bottleneck distance, which translates into a choice of $\tau$ that determines the final clustering. While AuToMATo is not parameter-free per se, we provide default choices that make it perform well across the board. Unless stated otherwise, AuToMATo will henceforth refer to our algorithm with its parameters set to these defaults. We experimentally analyze the clustering performance of AuToMATo and we find that it not only outperforms parameter-free clustering algorithms, but often also even the best choice of hyperparameters for many parametric clustering algorithms. Parameter-free algorithms building on ToMATo exist in the literature, for ex-

ample, in Cotsakis et al. (2021) the final clustering is determined by fitting a curve to the values of prominence, and in Bois et al. (2024) significant values are separated from non-significant ones by adapting the process that produces the persistence diagrams. Indeed, the former algorithm is one of those that AuToMATo is shown to outperform.

We envision one important application of AuToMATo to be to the *Mapper* algorithm, introduced in Singh et al. (2007). Mapper constructs a graph that captures the topological structure of a data set. It relies on many parameters, one them being a clustering algorithm applied to various chunks of the data. Algorithms that depend heavily on a a good choice of a tunable hyperparameter are generally not good candidates for usage with Mapper, as the best choice for the hyperparameter can vary significantly over the different chunks, and manually choosing a different hyperparameter for each may not be possible in practice. Thus, most choices of hyperparameter will generally perform badly on some of the subsets, leading to undesired results of Mapper. Thus AuToMATo can be seen as progress towards finding optimal parameters for Mapper, which is active area of research (Carrière et al., 2018; Chalapathi et al., 2021; Rosen et al., 2023). Running examples for Mapper with AuToMATo, we see that it is indeed a good choice for a clustering algorithm in this application when compared to parametric clustering algorithm such as DBSCAN.

## 2 BACKGROUND

### 2.1 PERSISTENCE AND THE TOMATO CLUSTERING ALGORITHM

Both ToMATo and AuToMATo rely on the theory of persistence (Edelsbrunner et al., 2002; Zomorodian & Carlsson, 2005; Carlsson, 2014) to quantify the prominence of peaks of (an estimate of) a density function, and to build a hierarchy of peaks. Given a topological space $X$ equipped with a density function $f\colon X \to \mathbb{R}_{\geq 0}$, the first step of persistence is to build a filtration from $X$.

**Definition 2.1.** *Let $X$ be a topological space and $f\colon X \to \mathbb{R}$ continuous. The **superlevelset filtration** of $(X, f)$ is the family of superlevelsets $\{X_{\geq t} \mid t \in \mathbb{R}\}$, where $X_{\geq t} := f^{-1}\left([t, \infty)\right)$.*

In the following we assume for ease of exposition that all local extrema of $f$ have distinct values. The idea underlying ToMATo is to track the evolution of (the number of) connected components of $X_{\geq t}$ as $t$ ranges from $+\infty$ to $-\infty$. In that process, the number of connected components of $X_{\geq t}$ remains constant, unless $t$ passes through the value of a local extremum of $f$. As $t$ passes through the value of a local maximum, a new connected component is "born" and added to the superlevelset $X_{\geq t}$. Similarly, as $t$ passes through the value of a local minimum, two connected components of $X_{\geq t}$ are merged into one. ToMATo builds a hierarchy of local maxima of $f$ by declaring that, as two components get merged, the component corresponding to the local maximum with higher value absorbs the other one and persists, whereas the component corresponding to the local maximum with lower value "dies". Therefore, to each local maximum we associate a pair $(b, d)$ where $b$ denotes the birth and $d$ the death time, respectively. The evolution of the connected components can be concisely recorded in a persistence diagram.

**Definition 2.2.** *Let $\{(b_l, d_l)\}_l$ denote the birth and death times of connected components of the superlevelset filtration $\{X_{\geq t}\}_{t \in \mathbb{R}}$ associated to the density $f\colon X \to \mathbb{R}$. The associated **persistence diagram**, denoted by $\mathrm{Dgm}(X, f)$, is the multiset in the extended plane $\overline{\mathbb{R}}^2 := \mathbb{R} \cup \{\pm\infty\}$ consisting of the points $\{(b_l, d_l)\}_l \subset \overline{\mathbb{R}}^2$ (counted with multiplicity) and the diagonal $\Delta := \left\{(x, x) \mid x \in \overline{\mathbb{R}}\right\}$ (where each point on $\Delta$ has infinite multiplicity). For a given local maximum of $f$ with birth time $b_l$ and death time $d_l$, we refer to the difference $d_l - b_l$ as its **prominence** or **lifetime**.*

The reason for working in the extended plane is that, provided that $f$ has a global maximum, the superlevelset filtration $X_{\geq t}$ will have a connected component that never dies, i.e., has death time equal to $-\infty$. See the red graph in Figure 1 for an illustration.

The persistence diagram $\mathrm{Dgm}(X, f)$ provides a summary of $f$. The points of $\mathrm{Dgm}(X, f)$ are in one-to-one correspondence with the local maxima of $f$, and twice the $L^\infty$-distance of a point to the diagonal $\Delta$ (i.e., its Euclidean vertical distance) equals its prominence.

We now outline how the ToMATo clustering algorithm works. Given a point cloud $X$ ToMATo relies on the assumption that the points of $X$ were sampled according to some unknown density function

$f$. In a nutshell, ToMATo infers information about the local maxima of $f$ by applying the above procedure to an estimate of $f$. ToMATo takes as input:

- **A neighborhood graph $\mathcal{G}$ on the points of $X$.** Chazal et al. mostly use the $\delta$-Rips graph and the $k$-nearest neighbor graph.[1]
- **A density estimator $\hat{f}$.** Each vertex $v$ of $\mathcal{G}$ is assigned a non-negative value $\hat{f}(v)$ that corresponds to the estimated density at $v$. Chazal et al. propose two possible density estimators: the truncated Gaussian kernel density estimator and the distance-to-measure density, originally introduced in Biau et al. (2011).[2]
- **A merging parameter $\tau \geq 0$.** This is a threshold that the prominence of a local maximum of the estimated density $\hat{f}$ must clear for that local maximum to be deemed a feature.

Given the inputs above, ToMATo proceeds as follows.

1. **Estimate the underlying density function $\hat{f}$ at the points of $X$.**
2. **Apply a hill-climbing algorithm on $\mathcal{G}$.** Construct the neighborhood graph $\mathcal{G}$ on the points of $X$, and construct a directed subgraph $\mathcal{G}'$ of $\mathcal{G}$ as follows: at each vertex $v$ of $\mathcal{G}$, place a directed edge from $v$ to its neighbor with highest value of $\hat{f}$, provided that that value is higher than $\hat{f}(v)$. If all neighbors of $v$ have lower values, $v$ is a peak of $\hat{f}$. This yields a collection of directed edges that form a spanning forest of the graph $\mathcal{G}$, consisting of one tree for each local maximum of $\hat{f}$. In particular, these trees yield a partition of the elements of $X$ into pairwise disjoint sets that serves as a candidate clustering on $X$.
3. **Construct the persistence diagram.** Construct the persistence diagram $\mathrm{Dgm}(\mathcal{G}, \hat{f})$ associated to the superlevelset filtration of $\hat{f} \colon \mathcal{G} \to \mathbb{R}$.
4. **Merge non-significant clusters.** Iteratively merge every cluster of prominence less than $\tau$ of the candidate clustering found in Step 2 into its parent cluster, i.e., into the cluster corresponding to the local maximum that it gets merged into in the superlevelset filtration of $\hat{f} \colon \mathcal{G} \to \mathbb{R}$. ToMATo outputs the resulting clustering of points of $X$, in which every cluster has prominence at least $\tau$ by construction.

The reason why we can expect the persistence diagram of the approximated density to be "close" to the original one stems from the stability of persistence diagrams under the bottleneck distance (explained in Section 2.2). This is illustrated in Figure 1.

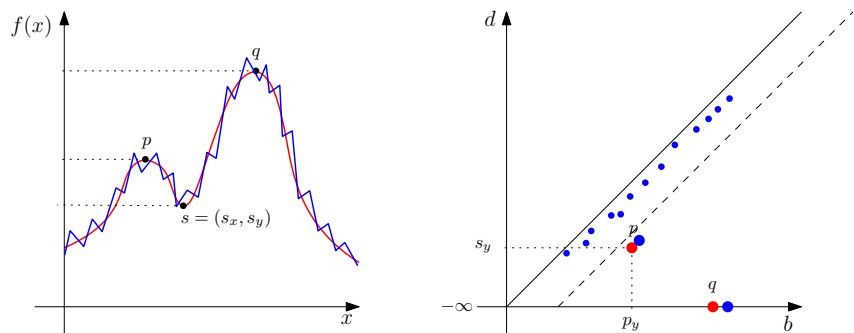

Figure 1: A function $f \colon K \to \mathbb{R}$, $K \subset \mathbb{R}$, in red, and an estimate $\hat{f}$ of $f$ in blue (left), with corresponding persistence diagrams $\mathrm{Dgm}(K, f)$ and $\mathrm{Dgm}(\mathcal{G}, \hat{f})$ consisting of the red and blue dots, respectively, together with a dashed line separating noise from features (right).

---

[1]Given a point cloud, both of these undirected graphs have the set of data points as their vertex set. In the case of the $\delta$-Rips graph, two vertices are connected iff they are at a distance of at most $\delta$ apart, whereas in the $k$-nearest neighbor graph, a data point is connected to another iff the latter is among the $k$-nearest neighbors of the first.

[2]For a smoothing parameter $m \in (0, 1)$, and a given data point $x$, its empirical (unnormalized) distance-to-measure density is given by $\hat{f}(x) = \left( \frac{1}{k} \sum_{y \in N_k(x)} \|x - y\|^2 \right)^{-\frac{1}{2}}$, where $k = \lceil mn \rceil$, $N_k(x)$ denotes the set of the $k$ nearest neighbors of $x$, and $n$ is the cardinality of the data set.

In practice, the user must run ToMATo twice. First, ToMATo is run with $\tau = +\infty$ which is equivalent to computing the birth and death time of each local maximum of $\hat{f}$ and hence the persistence diagram $\mathrm{Dgm}(\mathcal{G}, \hat{f})$. From the diagram $\mathrm{Dgm}(\mathcal{G}, \hat{f})$ the user then determines a merging parameter $\tau$ by visually identifying a large gap in $\mathrm{Dgm}(\mathcal{G}, \hat{f})$ separating, say, $C$ points corresponding to highly prominent peaks from the rest of the points. Then, ToMATo is run a second time with $\tau$ set to that value, which results in the final clustering of $X$ into $C$ clusters.

## 2.2 THE BOTTLENECK BOOTSTRAP

The bottleneck bootstrap, introduced in Chazal et al. (2017, Section 6), is used to separate significant features in persistence diagrams from non-significant ones. While it may be used in more general settings, we will restrict ourselves to the scenario of Section 2.1.

We first review the bottleneck distance, which is the standard distance measure between persistence diagrams (Edelsbrunner & Harer, 2010; Chazal et al., 2016).

**Definition 2.3.** *Let* $\mathrm{Dgm}_1$ *and* $\mathrm{Dgm}_2$ *be two persistence diagrams that have finitely many points off the diagonal. Let* $\pi$ *denote the set of bijections* $\nu\colon \mathrm{Dgm}_1 \to \mathrm{Dgm}_2$. *Given points* $x = (x_1, x_2)$ *and* $y = (y_1, y_2)$ *in* $\overline{\mathbb{R}}^2$, *let* $\|x - y\|_\infty = \max\{|x_1 - y_1|, |x_2 - y_2|\}$ *denote their* $L^\infty$-*distance, where we set* $(+\infty) - (+\infty) = (-\infty) - (-\infty) = 0$. *Then, the* **bottleneck distance** *between* $\mathrm{Dgm}_1$ *and* $\mathrm{Dgm}_2$ *is defined as*

$$W_\infty(\mathrm{Dgm}_1, \mathrm{Dgm}_2) = \inf_{\nu \in \pi} \sup_{x \in \mathrm{Dgm}_1} \|x - \nu(x)\|_\infty.$$

Note that a bijection $\nu\colon \mathrm{Dgm}_1 \to \mathrm{Dgm}_2$ is allowed to match an off-diagonal point of $\mathrm{Dgm}_1$ to the diagonal of $\mathrm{Dgm}_2$, and vice versa.

We now outline the bottleneck bootstrap. Suppose that $X$ is a sample consisting of $n$ data points, drawn according to some unknown probability density function $f\colon K \to [0, 1]$, $K \subset \mathbb{R}^n$, and let $\mathcal{D} := \mathrm{Dgm}(K, f)$ denote the corresponding (unknown) persistence diagram (we assume the density $f$ and all of its estimates to be normalized here for ease of exposition). We estimate $f$ and the connectivity of $K$ with a density estimator and a neighborhood graph, respectively (as explained in Section 2.1). This allows us to compute $\widehat{\mathcal{D}} := \mathrm{Dgm}(X, \hat{f})$ (where $\hat{f}$ is the estimate of $f$), which, in turn, serves as an estimate of $\mathcal{D}$.

Given a confidence level $\alpha \in (0, 1)$ and a number $B \in \mathbb{Z}_{\geq 1}$ of bootstrap iterations, the bottleneck bootstrap gives an estimate of $q_\alpha$, which is defined by

$$\mathbb{P}(\sqrt{n} W_\infty(\mathcal{D}, \widehat{\mathcal{D}}) \leq q_\alpha) = 1 - \alpha. \tag{1}$$

To that end, we first approximate $f$ with the empirical measure $P_n$ on $X$ that assigns the probability mass of $1/n$ to each data point (note that this generally does not coincide with $\hat{f}$). This allows us to estimate the distribution

$$F(z) := \mathbb{P}(\sqrt{n} W_\infty(\mathcal{D}, \widehat{\mathcal{D}}) \leq z)$$

with the distribution

$$\widehat{F}(z) := \mathbb{P}(\sqrt{n} W_\infty(\widehat{\mathcal{D}}^*, \widehat{\mathcal{D}}) \leq z),$$

where $\widehat{\mathcal{D}}^* := \mathrm{Dgm}(X^*, \hat{f}^*)$ is the persistence diagram corresponding to a sample $X^*$ of size $n$ drawn from $P_n$, and the density $\hat{f}^*$ and the connectivity of $X^*$ are estimated using the same estimators as before. Note that $X^*$ may be thought of as a sample drawn from $X$ with replacement. The distribution $\widehat{F}$ itself is approximated by Monte Carlo as follows. We draw $B$ samples $X_1^*, \dots, X_B^*$ of size $n$ from $P_n$, and for each of these $B$ samples, we compute the persistence diagram $\widehat{\mathcal{D}}_i^* := \mathrm{Dgm}(X_i^*, \hat{f}_i^*)$ and the quantity $T_i^* := \sqrt{n} W_\infty(\widehat{\mathcal{D}}_i^*, \widehat{\mathcal{D}})$, $i = 1, \dots, B$. Finally, we use the function

$$\widetilde{F}(z) := \frac{1}{B} \sum_{i=1}^{B} \mathbf{1}_{[0,z]}(T_i^*)$$

as an approximation of $\widehat{F}$, and hence of $F$. Using this, we set

$$\widehat{q}_\alpha := \inf\{z \mid \widetilde{F}(z) \geq 1 - \alpha\}$$

to be our estimate of $q_\alpha$. This estimate is asymptotically consistent if $\sup_z |\widetilde{F}(z) - F(z)| \xrightarrow{f} 0$. If that is the case, it follows from Equation 1 that (asymptotically) the true, unknown persistence diagram $\mathcal{D}$ is at bottleneck distance of at most $\widehat{q}_\alpha/\sqrt{n}$ from $\widehat{\mathcal{D}}$. Hence, points of $\widehat{\mathcal{D}}$ that are at $L^\infty$-distance at most $\widehat{q}_\alpha/\sqrt{n}$ from the diagonal could be matched to the diagonal under the bottleneck distance, and thus a point of $\widehat{\mathcal{D}}$ is declared to be a significant feature iff it is at $L^\infty$-distance of at least $\widehat{q}_\alpha/\sqrt{n}$ to the diagonal, i.e., iff its prominence is at least $2 \cdot \widehat{q}_\alpha/\sqrt{n}$.

## 3 METHODOLOGY AND IMPLEMENTATION OF AUTOMATO

### 3.1 METHODOLOGY OF AUTOMATO

AuToMATo builds upon the ToMATo clustering scheme introduced in Chazal et al. (2013) and implemented in Glisse (2023). AuToMATo automates the step of visual inspection of the persistence diagram by means of the bottleneck bootstrap, thus promoting ToMATo to a clustering scheme that does not rely on human input.

More precisely, given a point cloud $X$ to perform the clustering on, AuToMATo takes as input

- an instance of ToMATo with fixed neighborhood graph and density function estimators;
- a confidence level $\alpha \in (0, 1)$; and
- a number of bootstrap iterations $B \in \mathbb{Z}_{\geq 1}$.

**Remark 3.1.** *We point out that our implementation of AuToMATo comes with default values for each of the objects. Each of these values can, of course, be adjusted by the user. For details on these default values, see Subsection 3.2.*

In the present context the bottleneck bootstrap proceeds as follows. AuToMATo generates $B$ bootstrap subsamples $X_1^*, \ldots, X_B^*$ of $X$, each of the same cardinality as $X$. Then, the underlying ToMATo instance with $\tau = +\infty$ and its neighborhood graph and density function estimators is used to compute the persistence diagram for $X$ and each of $X_1^*, \ldots, X_B^*$, yielding persistence diagrams $\widehat{\mathcal{D}}$ and $\widehat{\mathcal{D}}_1^*, \ldots, \widehat{\mathcal{D}}_B^*$, respectively. Using the bootstrapped diagrams $\widehat{\mathcal{D}}_1^*, \ldots, \widehat{\mathcal{D}}_B^*$, a bottleneck bootstrap is performed on $\widehat{\mathcal{D}}$. This yields a value $\widehat{q}_\alpha$ that (asymptotically as $n \to \infty$) satisfies

$$\mathbb{P}(\sqrt{n}W_\infty(\mathcal{D}, \widehat{\mathcal{D}}) \leq \widehat{q}_\alpha) = 1 - \alpha,$$

where $\mathcal{D}$ denotes the persistence diagram of the true, unknown density function from which $X$ was sampled. Thus, points of $\widehat{\mathcal{D}}$ of prominence at least $2 \cdot \widehat{q}_\alpha/\sqrt{n}$ are declared to be significant features of $\widehat{\mathcal{D}}$, and AuToMATo outputs its underlying ToMATo instance with prominence threshold set to $\tau = 2 \cdot \widehat{q}_\alpha/\sqrt{n}$.

When computing the values $\sqrt{n}W_\infty(\widehat{\mathcal{D}}_i^*, \widehat{\mathcal{D}})$, $i = 1, \ldots, B$, in the bottleneck bootstrap, we only consider points in $\widehat{\mathcal{D}}_i^*$ and $\widehat{\mathcal{D}}$ with finite lifetimes. The reason for this choice is that we consider peaks with infinite lifetime to be significant a priori. Moreover, some of the bootstrapped diagrams among the $\widehat{\mathcal{D}}_1^*, \ldots, \widehat{\mathcal{D}}_B^*$ have a different number of points with infinite lifetime than the reference diagram $\widehat{\mathcal{D}}$. In these cases, the bottleneck distance of the bootstrapped diagram to the reference diagram is infinite, which heavily distorts the distribution $\widetilde{F}(z)$. This choice is justified by experiments.

### 3.2 IMPLEMENTATION OF AUTOMATO

We implemented AuToMATo in Python, and all of the code with documentation is available on GitHub.[3] For a description of AuToMATo in pseudocode, see Algorithm 1. The algorithm has a worst-case complexity of $O(B(nd + n\log(n) + N^{1.5}\log N))$, where $d$ is the dimensionality of the data and $N$ is the maximal number of off-diagonal points across all relevant persistence diagram (which is generally much smaller than $n$); see Appendix A.2 for details. Note that the factor of $B$ can be significantly decreased through parallelization.

While the input parameters may be adjusted by the user, the implementation provides default values whose choices we discuss presently.

---

[3]Anonymized GitHub-link

**Algorithm 1:** AuToMATo

---

**Input:** point cloud $X$ of $n$ data points; instance $\mathrm{tom}_\tau$ of ToMATo with neighborhood graph
and density function estimators, and prominence threshold $\tau$; confidence level
$\alpha \in (0,1)$; number of bootstrap iterations $B \in \mathbb{Z}_{\geq 1}$.

$\mathcal{D} \leftarrow \mathrm{Dgm}(\mathrm{tom}_\infty(X))$; `// compute persistence diagram of point cloud`
**for** $i = 1$ *to* $B$ **do**
  Let $X_i^*$ be a subsample of $X$ of size $n$, sampled with replacement;
  $\mathcal{D}_i^* \leftarrow \mathrm{Dgm}(\mathrm{tom}_\infty(X_i^*))$; `// compute persistence diagram of`
  `subsample`
  $d_i \leftarrow \sqrt{n} W_\infty^{\mathrm{fin}}(\mathcal{D}_i^*, \mathcal{D})$; `// compute bottleneck distance between`
  `finite points`
**end**
Sort and reindex $\{\mathcal{D}_1^*, \ldots, \mathcal{D}_B^*\}$ such that $d_1 \leq \cdots \leq d_B$;
$k \leftarrow \lceil (1 - \alpha) \cdot B \rceil$;
$\widehat{q}_\alpha \leftarrow d_k$;
$\tau \leftarrow 2 \cdot \widehat{q}_\alpha / \sqrt{n}$;

**Output:** $\mathrm{tom}_\tau(X)$; `// copy of initial ToMATo instance with`
`prominence threshold set to τ`

---

**Choice of ToMATo parameters:** Our implementation of AuToMATo is such that the user can directly pass parameters to the underlying ToMATo instance. If no such arguments are provided AuToMATo uses the default choices for those parameters, as determined by the implementation of ToMATo given in Glisse (2023). In particular, AuToMATo uses the $k$-nearest neighbor graph and the (logarithm of the) distance-to-measure density estimators by default, each with $k = 10$. Of course, the persistence diagrams produced by ToMATo, and hence the output of AuToMATo, depend on this choice. This can lead to suboptimal clustering performance of AuToMATo; see Section 6.

**Choice of $\alpha$ and $B$:** By default, AuToMATo performs the bootstrap on $B = 1000$ subsamples of the input point cloud, and sets the confidence level to $\alpha = 0.35$. The choice of this latter parameter means that AuToMATo determines merely a 65% confidence region for the persistence diagram produced by the underlying ToMATo instance. While in bootstrapping the confidence level is often set to e.g. $\alpha = 0.05$, the seemingly strange choice of $\alpha = 0.35$ in the setting of AuToMATo is justified by experiments. The value of 65% seems to be low enough to offset some of the negative influence of using possibly non-optimized neighborhood graph and density estimators discussed in Section 6, while at the same time being high enough to yield good results when these estimators are chosen suitably. We point out that the value $\alpha = 0.35$ (as well as the value $B = 1000$) was decided on after running an early implementation of AuToMATo on just a few synthetic data sets. In particular, the choice was made *before* conducting the experiments in Section 4. AuToMATo is implemented in such a way that the parameter $\alpha$ can be adjusted *after* fitting and the clustering is automatically updated.

Our Python package for AuToMATo consists of two separate modules; one for AuToMATo itself, and one for the bottleneck bootstrap. Both are compatible with the *scikit-learn* architecture, and the latter may also be used as a stand-alone module for other scenarios. In addition to the functionality inherited from the *scikit-learn* API, the implementation of AuToMATo comes with options of

- adjusting the parameter $\alpha$ of a fitted instance of AuToMATo which automatically updates the resulting clustering without repeating the (computationally expensive) bootstrapping;
- plotting the persistence diagram and the prominence threshold found in the bootstrapping;
- setting a seed in order to make the creation of the bootstrap subsamples in AuToMATo deterministic, thus allowing for reproducible results; and
- parallelizing the bottleneck bootstrap for speed improvements.

Finally, our implementation of AuToMATo contains a parameter that allows the algorithm to label points as outliers. In a nutshell, a point is classified as an outlier if it is not among the nearest

neighbors of more than a specified percentage of its own nearest neighbors. This feature, however, is currently experimental (and is thus turned off by default).

# 4 EXPERIMENTS

## 4.1 CHOICE OF CLUSTERING ALGORITHMS FOR COMPARISON

We chose to compare AuToMATo with its default parameters against

- DBSCAN and its extension HDBSCAN;
- hierarchical clustering with Ward, single, complete and average linkage;
- the FINCH clustering algorithm (Sarfraz et al., 2019); and
- a clustering algorithm building on ToMATo stemming from the *Topology ToolKit* (TTK) suite (Tierny et al., 2018); in the following, we will refer to this as the *TTK-algorithm*.[4]

For DBSCAN, HDBSCAN and the hierarchical clustering algorithms mentioned above, we worked with their implementations in *scikit-learn*.[5] For the FINCH clustering algorithm, we worked with the version available on GitHub.[6] Indeed, we subclassed that version in order to make it compatible with the *scikit-learn* API. Similarly, we created a *scikit-learn* compatible version of the TTK-algorithm by combining code from TTK with the description of the algorithm given in Cotsakis et al. (2021, Section 5.2). While we included DBSCAN and HDBSCAN among the clustering algorithms to compare AuToMATo against because they are standard choices, we chose to include the hierarchical clustering algorithms because they are readily available through *scikit-learn*. Finally, we chose to include FINCH and the TTK-algorithm because, like AuToMATo, they are out-of-the-box (indeed, parameter-free) methods and are thus especially interesting to compare AuToMATo against.

## 4.2 CHOICE OF DATA SETS

The data sets on which we ran AuToMATo and the above clustering algorithms stem from the *Clustering Benchmarks* suite.[7] We chose this collection as it comes with a large variety of different data sets, all of which are labeled by one or more ground truths, allowing for a fair and extensive comparison. The collection contains five recommended batteries of data sets from which we selected those (data set, ground truth)-pairs that we deemed reasonable for a general purpose parameter-free clustering algorithm. For instance, we chose to include the data set named `windows` that is part of the `wut`-battery, but not the data set named `windows` from the same battery (see Figure 5 in the appendix for an illustration). We chose to include the `windows` data set because AuToMATo determines clusters depending on connectivity, and topologically speaking, there is only one connected component in the `olympic` data set. Finally, we excluded all instances where the ground truth contains data points that are labeled as outliers, as outliers creation is currently an experimental feature in AuToMATo.

## 4.3 METHODOLOGY OF THE EXPERIMENTS

We min-max scaled each data set, fitted the clustering algorithms to them, and recorded the clustering performance of each result by computing the Fowlkes-Mallows score (Fowlkes & Mallows, 1983) of the clustering obtained and the respective ground truth. While the Fowlkes-Mallows score was originally defined for hierarchical clusterings only, it may be defined for general clusterings as follows. Given a clustering $C$ found by an algorithm and a ground truth clustering $G$, one defines the Fowlkes-Mallows score as

$$\text{FMS} := \sqrt{\frac{\text{TP}}{\text{TP} + \text{FP}}} \cdot \sqrt{\frac{\text{TP}}{\text{TP} + \text{FN}}},$$

where

- TP is the number of pairs of data points which are in the same cluster in $C$ and in $G$;

---

[4]For the *Topology ToolKit*, see `topology-tool-kit.github.io/` (BSD license).

[5]`scikit-learn.org/stable/modules/clustering.html`

[6]`github.com/ssarfraz/FINCH-Clustering` (CC BY-NC-SA 4.0 license)

[7]`clustering-benchmarks.gagolewski.com/` (CC BY-NC-ND 4.0 license)

- FP is the number of pairs of data points which are in the same cluster in $G$ but not in $C$; and

- FN is the number of pairs of data points which are not in the same cluster in $G$ but are in the same cluster in $C$.

In other words, the Fowlkes-Mallows score is defined as the geometric mean of precision and recall of a classifier whose relevant elements are pairs of points that belong to the same cluster in both $C$ and $G$. It may attain any value between 0 and 1, and these extremal values correspond to the worst and best possible clustering, respectively. We chose to use the Fowlkes-Mallows score as opposed to e.g. mutual information or any of the Rand indices, because the latter have been shown to exhibit biased behaviour depending on whether the clusters in the ground truth are mostly of similar sizes or not, see e.g. Romano et al. (2016); to the best of our knowledge, the Fowlkes-Mallows score does not suffer from such drawbacks. Moreover, we chose not to use any intrinsic measures of clustering performance since any such measure implicitly defines a further clustering algorithm to compare AuToMATo against, whereas we are interested in comparing AuToMATo against a predefined ground truth clustering.

We set the hyperparameters of the HDBSCAN, FINCH and the TTK-algorithm to their default values (as per their respective implementations). In contrast to this, we let the distance threshold parameter for the DBSCAN and the hierarchical clustering algorithms vary from 0.05 to 1.00 in increments of 0.05, with the goal of comparing AuToMATo against the best and worst performing version of these clustering algorithms. To account for the randomized component of AuToMATo, we ran it ten times, each time with a different seed.

While we restricted ourselves to instances where the ground truth does not contain any points labeled as outliers, some of the clustering algorithms in our list (DBSCAN and HDBSCAN) label some data points as outliers. In order to prevent these algorithms from getting systematically low Fowlkes-Mallows scores because of these outliers, we removed all the points labeled as outliers by these algorithms, and only computed the Fowlkes-Mallows score on the remaining points, both for these clustering algorithms and for AuToMATo. This of course gives an advantage to DBSCAN and HDBSCAN over AuToMATo.

In order to allow reproducibility, we chose a fixed seed for all our experiments, which can bey found in our code on GitHub. We ran our experiments on a laptop with a 12th Gen Intel Core i7-1260P processor running at 2.10GHz.

## 4.4 RESULTS AND INTERPRETATION

Table 1 shows the average Fowlkes-Mallows score of each algorithm across all benchmarking data sets; for AuToMATo, it shows the average and the standard deviation across the ten runs. For those benchmarking data sets that come with more than one ground truth, we included only the best score of the respective algorithm. Similarly, we included only the best performing parameter selection for those algorithms that we ran with varying distance thresholds (which, of course, skews the comparison in favor of those algorithms). As Table 1 shows, AuToMATo outperforms each clustering algorithm on average across all data sets, thus showing that it is indeed a versatile and powerful out-of-the-box clustering algorithm. In particular, AuToMATo outperforms the TTK-algorithm, which also build on ToMATo.

Table 1: Average clustering performance of AuToMATo vs. reference clustering algorithms

| Algorithm | Fowlkes-Mallows score |
|---|---|
| AuToMATo | **0.8554±0.0228** |
| DBSCAN | 0.8457 |
| Average linkage | 0.8321 |
| HDBSCAN | 0.8209 |
| Single linkage | 0.8156 |
| TTK clustering algorithm | 0.8019 |
| Complete linkage | 0.7592 |
| | Continued on next page |

Table 1: Average clustering performance of AuToMATo vs. reference clustering algorithms

| Algorithm | Fowlkes-Mallows score |
|---|---|
| Ward linkage | 0.5896 |
| FINCH | 0.5074 |

The scores of our experiments are reported in Tables 2 through 6 in Appendix A.3. As an illustration, Figure 2 shows that the best choice of parameter for DBSCAN sometimes outperforms AuToMATo, which is to be expected. However, on most data sets where this is the case, the results from Au-ToMATo are still competitive, and there is a significant number of instances where AuToMATo outperforms DBSCAN for all parameter selections, in some cases by a lot.

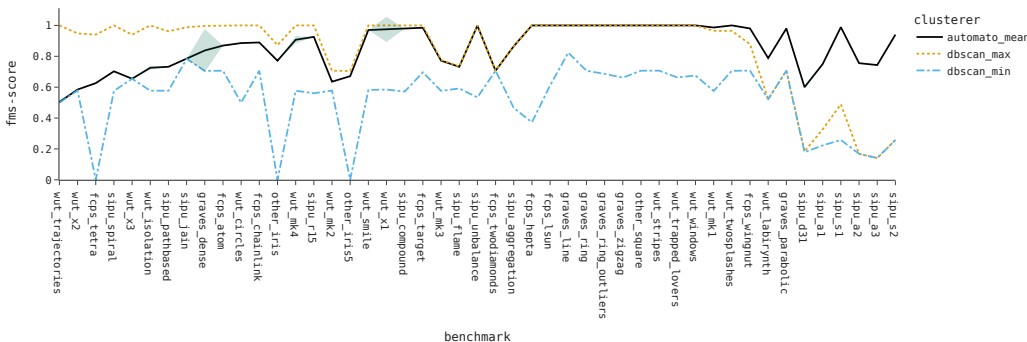

Figure 2: Fowlkes-Mallows score of AuToMATo and DBSCAN across benchmarking data sets. The shading of "automato_mean" indicates the standard deviation of the score across the ten runs.

## 5 APPLICATIONS OF AUTOMATO IN COMBINATION WITH MAPPER

The goal of *Mapper* (Singh et al., 2007) is to approximate the *Reeb graph* of a manifold $M$ based on a sample from $M$. The input is a point cloud $P$ with a filter function $P \to \mathbb{R}$, a collection of overlapping intervals $\mathcal{U} = \{U_1, \ldots, U_n\}$ covering $\mathbb{R}$ and a clustering algorithm. For each $U_i \in \mathcal{U}$, Mapper runs the clustering algorithm on the data points in the preimage $f^{-1}(U_i)$, creating a vertex for each cluster. Two vertices are then connected by an edge if the corresponding clusters (in different preimages) have some data points in common, yielding a graph that represents the shape of the data set. We ran the Mapper implementation of *giotto-tda* (Tauzin et al., 2020) on a synthetic two-dimensional data set consisting of noisy samples from two concentric circles (see Figure 3a) with projection onto the $x$-axis as the filter function. We ran Mapper on the same interval cover with three different choices of clustering algorithms: AuToMATo, DBSCAN, and HDBSCAN. As can be seen in Figure 3b, using DBSCAN, we get many unwanted edges in the graph. HDBSCAN performs better, giving two cycles with some extra loops. The output of Mapper with AuToMATo is exactly the Reeb graph of two circles.

We further tested the combination of Mapper with AuToMATo on one of the standard applications of Mapper: the Miller-Reaven diabetes data set, where Mapper can be used detect two strains of diabetes that correspond to "flares" in the data set (see Singh et al. (2007, Section 5.1) for details).[8] As can be seen in Figure 4, AuToMATo performs well in this task; the graphs show a central core of vertices corresponding to healthy patients, and two flares corresponding to the two strains of diabetes. We were not able to reproduce this using DBSCAN or HDBSCAN; Figure 4 shows the output of Mapper with these algorithms with their respective default parameters.

---

[8]The data set is available as part of the "locfit" R-package (Loader, 2024).

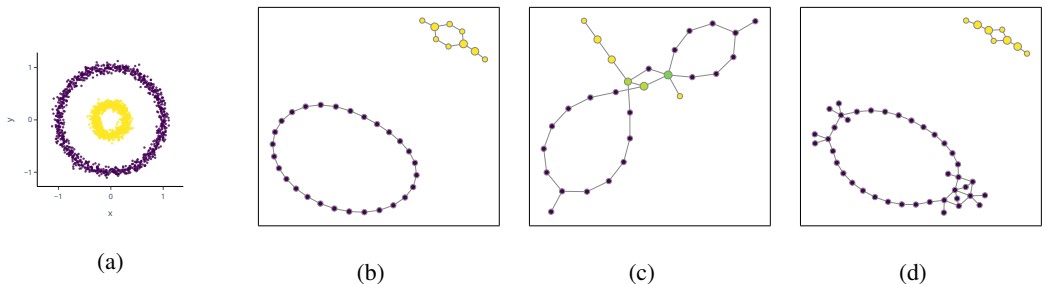

Figure 3: (a) input data set; result of Mapper with (b) AuToMATo; (c) DBSCAN; (d) HDBSCAN

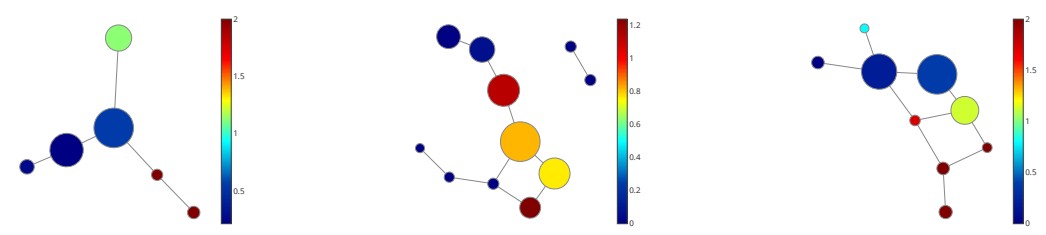

Figure 4: Mapper applied to the diabetes data set with AuToMATo (left); DBSCAN (center); HDBSCAN (right). Labels 0, 1 and 2 stand for "no ", "chemical" and "overt diabetes".

## 6  DISCUSSION

We briefly outline some limitations of AuToMATo. AuToMATo comes with a choice of default values for its parameters. In particular, it resorts to the default values as implemented in ToMATo for the choice of neighborhood graph and density estimators. In ToMATo, the options for the neighborhood graph estimators are the $\delta$-Rips graph and the $k$-nearest neighbor graph, relying on parameters $\delta > 0$ and $k \geq 1$, respectively; the options for the density estimators are the kernel density estimator and the distance-to-measure density estimator, which in turn rely parameters $h > 0$ and $m \in (0, 1)$, respectively (see the footnotes in Section 2.1 for their definitions). While for most data sets that we ran our experiments on these default estimators yielded good results, there are cases where adjusting them before running AuToMATo improves the clustering performance. Nevertheless, since finding a priori optimal parameters for the neighborhood graph and density estimators for a given data set is largely an open problem, we chose to stick to the default values for these estimators. Indeed, there are heuristics for the selection of the bandwidth in kernel density estimation (see Cotsakis et al. (2021, Section 4.1)) and the smoothing parameter in distance-to-measure density estimation (see Chazal et al. (2017, Section 7.1)), but these methods either lack theoretical justification, or would require running AuToMATo multiple times on the same data set with different parameters and selecting the best ones a posteriori, which undermines the intended use of AuToMATo as an out-of-the-box clustering algorithm.

Optimizing the choice of the neighborhood graph and density estimators is an aspect of AuToMATo that we plan to pursue in future work. Moreover, we plan to improve the currently experimental feature for outlier creation in AuToMATo discussed at the end of Section 3.2. Finally, it is natural to ask whether the results from Carrière et al. (2018) on optimal parameter selection in the Mapper algorithm can be adapted to the scenario where Mapper uses AuToMATo as its clustering algorithm.

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

# A  APPENDIX

## A.1  ABOUT THE CHOICE OF DATA SETS

As explained in Section 4.2, we chose to include the data set named `windows` from the battery named `wut`, but not the data set named `olympic` from the same battery. Those are illustrated in Figure 5. In that figure, the data points are colored according to the ground truth clustering.

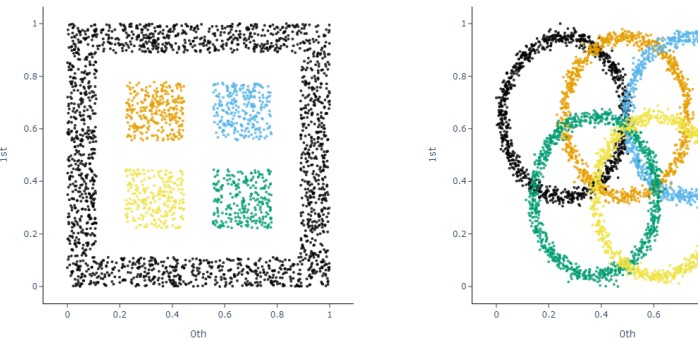

Figure 5: The data sets named `windows` (left) and `olympic` (right) from the `wut`-battery.

## A.2  COMPLEXITY ANALYSIS OF ALGORITHM 1

Recall from Chazal et al. (2013, Section 2) that, if an estimated density and a neighborhood graph are provided, ToMATo has a worst-case time complexity in $O(n \log(n) + m\alpha(n))$, where $n$ and $m$ are the number of vertices and edges of the neighborhood graph, respectively, and $\alpha$ denotes the inverse Ackermann function (note that $n$ equals the number of data points). By default, ToMATo (and hence AuToMATo) works with the $k$-nearest neighbor graph and distance-to-measure density estimators, where the latter relies itself on the $k$-nearest neighbor graph (each with $k = 10$). Taking into account the known complexity bound $O(nd)$ for the creation of the $k$-nearest neighbor graph (where $d$ is the dimensionality of the data), and using the fact that $m \in O(n)$ for this graph, this leads to a worst-case time complexity in $O(nd + n \log(n))$ for a single run of ToMATo. Creating the bootstrap samples $X_i^*$, $i = 1, \ldots, B$, has complexity in $O(Bn)$; computing the values $\sqrt{n} W_\infty^{\text{fin}}(\mathcal{D}_i^*, \mathcal{D})$, $i = 1, \ldots, B$, has worst-case complexity $O(BN^{1.5} \log(N))$ (see e.g. Efrat et al. (2001); here $N$ denotes the maximal number of off-diagonal points across all relevant persistence diagram), and sorting them has worst case complexity in $O(B \log(B))$. Combined, this leads to a worst-case complexity for AuToMATo in $O(B(nd + n \log(n) + N^{1.5} \log(N)) + B \log(B))$. Using that $B$ is a constant, we obtain the runtime claimed in the main body.

## A.3  BENCHMARKING RESULTS

In this subsection we report the Fowlkes-Mallows scores coming from comparing AuToMATo to the other clustering algorithms, as explained in Section 4. For those benchmarking data sets that come with more than one ground truth, we report the scores for each of those, and different ground truths are indicated by the last digit in the data set name. Moreover, each table is sorted according to increasing difference in clustering performance of AuToMATo and the respective clustering algorithm that AuToMATo is being compared against. As is customary, we indicate the score stemming from the best performing clustering algorithm in bold. Finally, each of the table is accompanied by a graph similar to the one depicted in Figure 2. Note that, in particular, that those figures indicate only the score corresponding to the ground truth on which the respective clustering algorithm performs best on.

Table 2: Fowlkes-Mallows scores of AuToMATo vs. DBSCAN

| Dataset | automato_mean | dbscan_max | dbscan_min |
| --- | --- | --- | --- |
| sipu_r15_2 | 0.4867±0.0000 | **1.0000** | 0.5607 |
| wut_trajectories_0 | 0.5038±0.0107 | **1.0000** | 0.4999 |
| wut_x3_0 | 0.5153±0.0000 | **0.9398** | 0.5149 |
| wut_x2_0 | 0.5846±0.0000 | **0.9483** | 0.5779 |
| sipu_r15_1 | 0.5436±0.0000 | **0.8954** | 0.5021 |
| fcps_tetra_0 | 0.6261±0.0000 | **0.9403** | 0.0000 |
| sipu_pathbased_0 | 0.6517±0.0000 | **0.9569** | 0.5769 |
| sipu_spiral_0 | 0.7028±0.0000 | **1.0000** | 0.5756 |
| wut_isolation_0 | 0.7256±0.0113 | **1.0000** | 0.5773 |
| sipu_pathbased_1 | 0.7322±0.0000 | **0.9620** | 0.5170 |
| sipu_jain_0 | 0.7837±0.0000 | **0.9880** | 0.7837 |
| graves_dense_0 | 0.8377±0.1396 | **0.9970** | 0.7053 |
| sipu_compound_0 | 0.8616±0.0000 | **1.0000** | 0.4972 |
| fcps_atom_0 | 0.8694±0.0000 | **1.0000** | 0.7067 |
| wut_circles_0 | 0.8857±0.0000 | **1.0000** | 0.4998 |
| fcps_chainlink_0 | 0.8896±0.0000 | **1.0000** | 0.7068 |
| other_iris_0 | 0.7715±0.0000 | **0.8721** | 0.0000 |
| wut_mk4_0 | 0.9072±0.0234 | **1.0000** | 0.5770 |
| wut_mk2_0 | 0.6356±0.0000 | **0.7068** | 0.5778 |
| sipu_compound_4 | 0.9442±0.0000 | **1.0000** | 0.5523 |
| wut_x3_1 | 0.6546±0.0000 | **0.7042** | 0.6546 |
| graves_zigzag_1 | 0.6720±0.0000 | **0.7149** | 0.4446 |
| other_iris5_0 | 0.6712±0.0000 | **0.7046** | 0.0000 |
| wut_smile_1 | 0.9701±0.0000 | **1.0000** | 0.5825 |
| wut_x1_0 | 0.9741±0.0818 | **1.0000** | 0.5846 |
| fcps_target_0 | 0.9850±0.0000 | **1.0000** | 0.6963 |
| wut_smile_0 | 0.9681±0.0000 | **0.9753** | 0.5471 |
| wut_mk3_0 | 0.7720±0.0000 | **0.7774** | 0.5764 |
| sipu_compound_1 | 0.9786±0.0000 | **0.9825** | 0.5715 |
| sipu_flame_0 | 0.7320±0.0000 | **0.7341** | 0.5918 |
| sipu_unbalance_0 | 0.9986±0.0008 | **1.0000** | 0.5339 |
| fcps_twodiamonds_0 | **0.7067±0.0000** | **0.7067** | **0.7067** |
| sipu_aggregation_0 | **0.8652±0.0000** | **0.8652** | 0.4653 |
| wut_stripes_0 | **1.0000±0.0000** | **1.0000** | 0.7070 |
| wut_trapped_lovers_0 | **1.0000±0.0000** | **1.0000** | 0.6632 |
| wut_windows_0 | **1.0000±0.0000** | **1.0000** | 0.6753 |
| fcps_hepta_0 | **1.0000±0.0000** | **1.0000** | 0.3727 |
| fcps_lsun_0 | **1.0000±0.0000** | **1.0000** | 0.6111 |
| graves_line_0 | **1.0000±0.0000** | **1.0000** | 0.8238 |
| graves_ring_0 | **1.0000±0.0000** | **1.0000** | 0.7068 |
| graves_ring_outliers_0 | **1.0000±0.0000** | **1.0000** | 0.6863 |
| graves_zigzag_0 | **1.0000±0.0000** | **1.0000** | 0.5328 |
| other_square_0 | **1.0000±0.0000** | **1.0000** | 0.7068 |
| wut_mk1_0 | **0.9866±0.0000** | 0.9651 | 0.5754 |
| wut_twosplashes_0 | **1.0000±0.0000** | 0.9649 | 0.7062 |
| fcps_wingnut_0 | **0.9805±0.0000** | 0.8784 | 0.7068 |
| graves_parabolic_1 | **0.6916±0.0000** | 0.5000 | 0.4999 |
| wut_labirynth_0 | **0.7884±0.0000** | 0.5221 | 0.5221 |
| graves_parabolic_0 | **0.9802±0.0000** | 0.7068 | 0.7068 |
| sipu_d31_0 | **0.6001±0.0085** | 0.1846 | 0.1787 |
| sipu_a1_0 | **0.7499±0.0000** | 0.3269 | 0.2229 |
| sipu_r15_0 | **0.9258±0.0000** | 0.4551 | 0.2552 |
| sipu_s1_0 | **0.9888±0.0000** | 0.4890 | 0.2581 |

Table 2: Fowlkes-Mallows scores of AuToMATo vs. DBSCAN

| Dataset | automato_mean | dbscan_max | dbscan_min |
|---------|---------------|------------|------------|
| sipu_a2_0 | **0.7555±0.0000** | 0.1685 | 0.1685 |
| sipu_a3_0 | **0.7434±0.0000** | 0.1410 | 0.1410 |
| sipu_s2_0 | **0.9405±0.0000** | 0.2581 | 0.2581 |

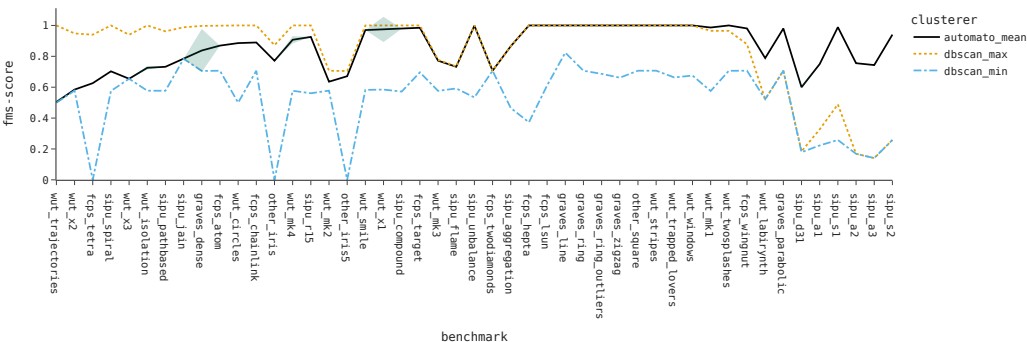

Figure 6: Comparison of AuToMATo and DBSCAN.

Table 3: Fowlkes-Mallows scores of AuToMATo vs. hierarchical clustering with average linkage

| Dataset | automato_mean | linkage_average_max | linkage_average_min |
|---------|---------------|---------------------|---------------------|
| sipu_r15_2 | 0.4867±0.0000 | **1.0000** | 0.3971 |
| wut_trajectories_0 | 0.5038±0.0107 | **1.0000** | 0.3115 |
| fcps_tetra_0 | 0.6261±0.0000 | **1.0000** | 0.0651 |
| sipu_r15_1 | 0.5436±0.0000 | **0.8954** | 0.4435 |
| sipu_d31_0 | 0.6001±0.0085 | **0.9322** | 0.1787 |
| wut_x3_0 | 0.5153±0.0000 | **0.8389** | 0.3343 |
| wut_x3_1 | 0.6546±0.0000 | **0.9747** | 0.2693 |
| fcps_twodiamonds_0 | 0.7067±0.0000 | **0.9925** | 0.1287 |
| sipu_a2_0 | 0.7555±0.0000 | **0.9432** | 0.1685 |
| sipu_a1_0 | 0.7499±0.0000 | **0.9268** | 0.2229 |
| sipu_a3_0 | 0.7434±0.0000 | **0.8825** | 0.1410 |
| sipu_aggregation_0 | 0.8652±0.0000 | **0.9932** | 0.1785 |
| sipu_pathbased_1 | 0.7322±0.0000 | **0.8564** | 0.2058 |
| graves_dense_0 | 0.8377±0.1396 | **0.9604** | 0.6633 |
| sipu_pathbased_0 | 0.6517±0.0000 | **0.7704** | 0.1848 |
| wut_x2_0 | 0.5846±0.0000 | **0.7001** | 0.2782 |
| wut_circles_0 | 0.8857±0.0000 | **1.0000** | 0.2369 |
| wut_mk3_0 | 0.7720±0.0000 | **0.8771** | 0.0876 |
| wut_mk2_0 | 0.6356±0.0000 | **0.7068** | 0.1421 |
| sipu_r15_0 | 0.9258±0.0000 | **0.9900** | 0.2552 |
| graves_zigzag_1 | 0.6720±0.0000 | **0.7202** | 0.4380 |
| other_iris_0 | 0.7715±0.0000 | **0.8080** | 0.0791 |
| other_iris5_0 | 0.6712±0.0000 | **0.7042** | 0.0451 |
| wut_x1_0 | 0.9741±0.0818 | **1.0000** | 0.3070 |
| sipu_jain_0 | 0.7837±0.0000 | **0.7904** | 0.1736 |
| wut_mk1_0 | 0.9866±0.0000 | **0.9933** | 0.2037 |
| sipu_unbalance_0 | 0.9986±0.0008 | **0.9995** | 0.5339 |
| fcps_hepta_0 | **1.0000±0.0000** | 1.0000 | 0.3727 |

Table 3: Fowlkes-Mallows scores of AuToMATo vs. hierarchical clustering with average linkage

| Dataset | automato_mean | linkage_average_max | linkage_average_min |
|---|---|---|---|
| sipu_flame_0 | **0.7320±0.0000** | **0.7320** | 0.0913 |
| sipu_s1_0 | **0.9888±0.0000** | 0.9821 | 0.2581 |
| sipu_compound_0 | **0.8616±0.0000** | 0.8431 | 0.2207 |
| sipu_compound_4 | **0.9442±0.0000** | 0.9224 | 0.1985 |
| sipu_compound_1 | **0.9786±0.0000** | 0.9546 | 0.1922 |
| sipu_s2_0 | **0.9405±0.0000** | 0.9097 | 0.2581 |
| wut_smile_1 | **0.9701±0.0000** | 0.8726 | 0.4041 |
| fcps_atom_0 | **0.8694±0.0000** | 0.7491 | 0.2555 |
| graves_parabolic_1 | **0.6916±0.0000** | 0.5708 | 0.2135 |
| sipu_spiral_0 | **0.7028±0.0000** | 0.5756 | 0.1919 |
| wut_mk4_0 | **0.9072±0.0234** | 0.7714 | 0.2071 |
| wut_smile_0 | **0.9681±0.0000** | 0.8221 | 0.4303 |
| wut_isolation_0 | **0.7256±0.0113** | 0.5773 | 0.1651 |
| graves_line_0 | **1.0000±0.0000** | 0.8238 | 0.3047 |
| fcps_chainlink_0 | **0.8896±0.0000** | 0.7068 | 0.1456 |
| fcps_target_0 | **0.9850±0.0000** | 0.7986 | 0.3285 |
| fcps_wingnut_0 | **0.9805±0.0000** | 0.7739 | 0.1101 |
| fcps_lsun_0 | **1.0000±0.0000** | 0.7896 | 0.1735 |
| graves_ring_0 | **1.0000±0.0000** | 0.7780 | 0.2638 |
| graves_parabolic_0 | **0.9802±0.0000** | 0.7580 | 0.1598 |
| graves_ring_outliers_0 | **1.0000±0.0000** | 0.7767 | 0.2801 |
| other_square_0 | **1.0000±0.0000** | 0.7413 | 0.1746 |
| wut_labirynth_0 | **0.7884±0.0000** | 0.5221 | 0.2306 |
| wut_stripes_0 | **1.0000±0.0000** | 0.7070 | 0.1082 |
| wut_twosplashes_0 | **1.0000±0.0000** | 0.7062 | 0.4837 |
| wut_windows_0 | **1.0000±0.0000** | 0.6753 | 0.1194 |
| wut_trapped_lovers_0 | **1.0000±0.0000** | 0.6632 | 0.1077 |
| graves_zigzag_0 | **1.0000±0.0000** | 0.6616 | 0.3008 |

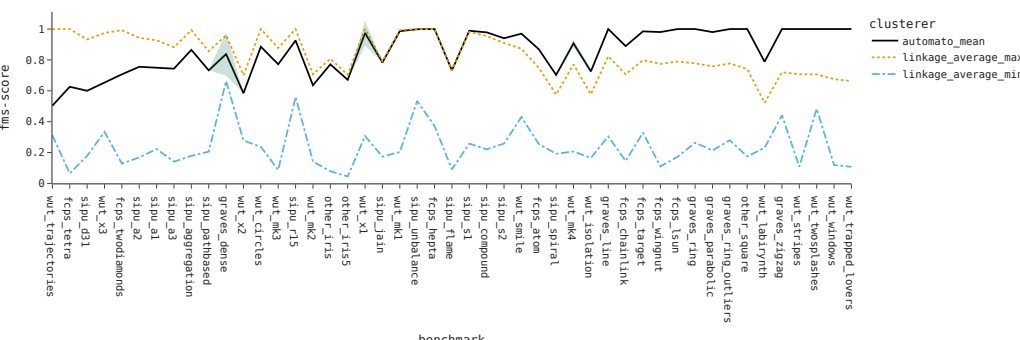

Figure 7: Comparison of AuToMATo and agglomerative clustering with average linkage.

Table 4: Fowlkes-Mallows scores of AuToMATo vs. HDBSCAN

| Dataset | automato_mean | automato_std | hdbscan |
|---|---|---|---|
| wut_trajectories_0 | 0.5038±0.0107 | 0.0107 | **1.0000** |
| wut_x3_0 | 0.5153±0.0000 | 0.0000 | **0.8959** |
| wut_x2_0 | 0.5846±0.0000 | 0.0000 | **0.9344** |

Table 4: Fowlkes-Mallows scores of AuToMATo vs. HDBSCAN

| Dataset | automato_mean | automato_std | hdbscan |
|---|---|---|---|
| sipu_spiral_0 | 0.7028±0.0000 | 0.0000 | **0.9815** |
| sipu_flame_0 | 0.7320±0.0000 | 0.0000 | **0.9900** |
| sipu_d31_0 | 0.6001±0.0085 | 0.0085 | **0.8231** |
| sipu_jain_0 | 0.7837±0.0000 | 0.0000 | **0.9779** |
| fcps_tetra_0 | 0.6261±0.0000 | 0.0000 | **0.8157** |
| graves_dense_0 | 0.8377±0.1396 | 0.1396 | **0.9894** |
| sipu_pathbased_1 | 0.7322±0.0000 | 0.0000 | **0.8634** |
| fcps_atom_0 | 0.8694±0.0000 | 0.0000 | **1.0000** |
| sipu_pathbased_0 | 0.6517±0.0000 | 0.0000 | **0.7815** |
| fcps_chainlink_0 | 0.8896±0.0000 | 0.0000 | **1.0000** |
| sipu_r15_0 | 0.9258±0.0000 | 0.0000 | **0.9932** |
| sipu_a1_0 | 0.7499±0.0000 | 0.0000 | **0.8081** |
| wut_x3_1 | 0.6546±0.0000 | 0.0000 | **0.6972** |
| other_iris5_0 | 0.6712±0.0000 | 0.0000 | **0.7042** |
| wut_x1_0 | 0.9741±0.0818 | 0.0818 | **1.0000** |
| sipu_compound_1 | 0.9786±0.0000 | 0.0000 | **1.0000** |
| sipu_compound_4 | 0.9442±0.0000 | 0.0000 | **0.9656** |
| fcps_target_0 | 0.9850±0.0000 | 0.0000 | **1.0000** |
| sipu_compound_0 | 0.8616±0.0000 | 0.0000 | **0.8751** |
| sipu_unbalance_0 | 0.9986±0.0008 | 0.0008 | **1.0000** |
| graves_zigzag_1 | **0.6720±0.0000** | 0.0000 | **0.6720** |
| sipu_aggregation_0 | **0.8652±0.0000** | 0.0000 | **0.8652** |
| wut_stripes_0 | **1.0000±0.0000** | 0.0000 | **1.0000** |
| wut_trapped_lovers_0 | **1.0000±0.0000** | 0.0000 | **1.0000** |
| wut_windows_0 | **1.0000±0.0000** | 0.0000 | **1.0000** |
| fcps_hepta_0 | **1.0000±0.0000** | 0.0000 | **1.0000** |
| fcps_lsun_0 | **1.0000±0.0000** | 0.0000 | **1.0000** |
| graves_line_0 | **1.0000±0.0000** | 0.0000 | **1.0000** |
| graves_ring_0 | **1.0000±0.0000** | 0.0000 | **1.0000** |
| graves_ring_outliers_0 | **1.0000±0.0000** | 0.0000 | **1.0000** |
| graves_zigzag_0 | **1.0000±0.0000** | 0.0000 | **1.0000** |
| other_square_0 | **1.0000±0.0000** | 0.0000 | **1.0000** |
| other_iris_0 | **0.7715±0.0000** | 0.0000 | **0.7715** |
| wut_mk3_0 | **0.7720±0.0000** | 0.0000 | 0.7719 |
| wut_mk1_0 | **0.9866±0.0000** | 0.0000 | 0.9863 |
| sipu_a3_0 | **0.7434±0.0000** | 0.0000 | 0.7415 |
| sipu_a2_0 | **0.7555±0.0000** | 0.0000 | 0.7502 |
| sipu_r15_2 | **0.4867±0.0000** | 0.0000 | 0.4671 |
| sipu_r15_1 | **0.5436±0.0000** | 0.0000 | 0.5212 |
| wut_isolation_0 | **0.7256±0.0113** | 0.0113 | 0.6377 |
| fcps_wingnut_0 | **0.9805±0.0000** | 0.0000 | 0.8725 |
| sipu_s1_0 | **0.9888±0.0000** | 0.0000 | 0.8717 |
| sipu_s2_0 | **0.9405±0.0000** | 0.0000 | 0.7410 |
| wut_mk4_0 | **0.9072±0.0234** | 0.0234 | 0.6459 |
| wut_labirynth_0 | **0.7884±0.0000** | 0.0000 | 0.5134 |
| graves_parabolic_1 | **0.6916±0.0000** | 0.0000 | 0.3616 |
| fcps_twodiamonds_0 | **0.7067±0.0000** | 0.0000 | 0.2886 |
| wut_mk2_0 | **0.6356±0.0000** | 0.0000 | 0.1574 |
| wut_smile_0 | **0.9681±0.0000** | 0.0000 | 0.4000 |
| wut_smile_1 | **0.9701±0.0000** | 0.0000 | 0.3714 |
| graves_parabolic_0 | **0.9802±0.0000** | 0.0000 | 0.3526 |
| wut_twosplashes_0 | **1.0000±0.0000** | 0.0000 | 0.3074 |
| wut_circles_0 | **0.8857±0.0000** | 0.0000 | 0.1204 |

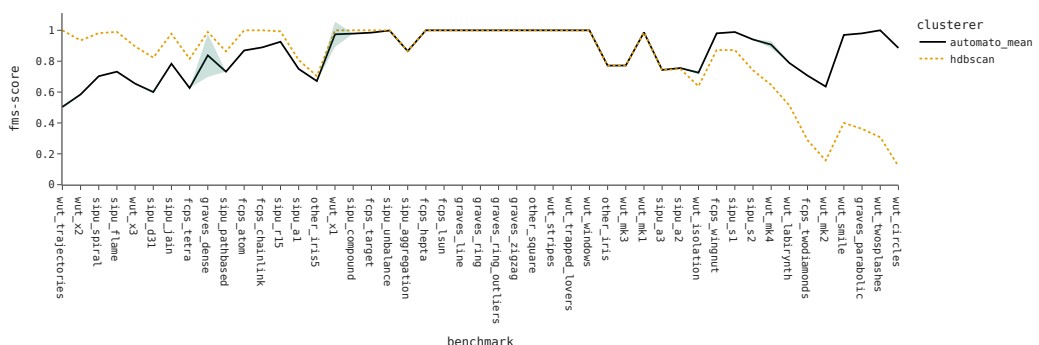

Figure 8: Comparison of AuToMATo and HDBSCAN.

Table 5: Fowlkes-Mallows scores of AuToMATo vs. hierarchical clustering with single linkage

| Dataset | automato_mean | linkage_single_max | linkage_single_min |
| --- | --- | --- | --- |
| sipu_r15_2 | 0.4867±0.0000 | **1.0000** | 0.5607 |
| wut_trajectories_0 | 0.5038±0.0107 | **1.0000** | 0.4999 |
| sipu_r15_1 | 0.5436±0.0000 | **0.8954** | 0.5021 |
| fcps_tetra_0 | 0.6261±0.0000 | **0.9296** | 0.0829 |
| sipu_spiral_0 | 0.7028±0.0000 | **1.0000** | 0.5756 |
| wut_isolation_0 | 0.7256±0.0113 | **1.0000** | 0.5773 |
| wut_x3_0 | 0.5153±0.0000 | **0.7347** | 0.4951 |
| sipu_jain_0 | 0.7837±0.0000 | **0.9510** | 0.7837 |
| fcps_atom_0 | 0.8694±0.0000 | **1.0000** | 0.7067 |
| wut_circles_0 | 0.8857±0.0000 | **1.0000** | 0.4998 |
| fcps_chainlink_0 | 0.8896±0.0000 | **1.0000** | 0.7068 |
| wut_mk4_0 | 0.9072±0.0234 | **1.0000** | 0.5770 |
| sipu_compound_0 | 0.8616±0.0000 | **0.9454** | 0.4972 |
| sipu_pathbased_0 | 0.6517±0.0000 | **0.7337** | 0.5769 |
| sipu_pathbased_1 | 0.7322±0.0000 | **0.8091** | 0.5170 |
| graves_dense_0 | 0.8377±0.1396 | **0.9096** | 0.6882 |
| wut_mk2_0 | 0.6356±0.0000 | **0.7068** | 0.6007 |
| graves_zigzag_1 | 0.6720±0.0000 | **0.7344** | 0.4446 |
| wut_x2_0 | 0.5846±0.0000 | **0.6437** | 0.5105 |
| other_iris5_0 | 0.6712±0.0000 | **0.7042** | 0.0451 |
| wut_smile_1 | 0.9701±0.0000 | **1.0000** | 0.5825 |
| wut_x1_0 | 0.9741±0.0818 | **0.9920** | 0.5846 |
| fcps_target_0 | 0.9850±0.0000 | **1.0000** | 0.6963 |
| wut_smile_0 | 0.9681±0.0000 | **0.9748** | 0.5471 |
| sipu_unbalance_0 | 0.9986±0.0008 | **1.0000** | 0.5339 |
| fcps_twodiamonds_0 | **0.7067±0.0000** | 0.7067 | **0.7067** |
| wut_x3_1 | **0.6546±0.0000** | 0.6546 | 0.6140 |
| sipu_aggregation_0 | **0.8652±0.0000** | 0.8652 | 0.4653 |
| wut_stripes_0 | **1.0000±0.0000** | 1.0000 | 0.7070 |
| wut_trapped_lovers_0 | **1.0000±0.0000** | 1.0000 | 0.6632 |
| wut_windows_0 | **1.0000±0.0000** | 1.0000 | 0.6753 |
| fcps_hepta_0 | **1.0000±0.0000** | 1.0000 | 0.3727 |
| graves_line_0 | **1.0000±0.0000** | 1.0000 | 0.8238 |
| graves_ring_0 | **1.0000±0.0000** | 1.0000 | 0.7068 |
| graves_ring_outliers_0 | **1.0000±0.0000** | 1.0000 | 0.6863 |
| graves_zigzag_0 | **1.0000±0.0000** | 1.0000 | 0.5381 |
| | | | Continued on next page |

Table 5: Fowlkes-Mallows scores of AuToMATo vs. hierarchical clustering with single linkage

| Dataset | automato_mean | linkage_single_max | linkage_single_min |
|---------|---------------|--------------------|--------------------|
| sipu_flame_0 | **0.7320±0.0000** | **0.7320** | 0.4598 |
| other_iris_0 | **0.7715±0.0000** | **0.7715** | 0.1223 |
| other_square_0 | **1.0000±0.0000** | 0.9990 | 0.7068 |
| fcps_lsun_0 | **1.0000±0.0000** | 0.9983 | 0.6111 |
| wut_twosplashes_0 | **1.0000±0.0000** | 0.9850 | 0.7062 |
| sipu_compound_1 | **0.9786±0.0000** | 0.9180 | 0.5715 |
| sipu_compound_4 | **0.9442±0.0000** | 0.8824 | 0.5523 |
| wut_mk1_0 | **0.9866±0.0000** | 0.8866 | 0.5754 |
| fcps_wingnut_0 | **0.9805±0.0000** | 0.8087 | 0.7068 |
| graves_parabolic_1 | **0.6916±0.0000** | 0.5000 | 0.4979 |
| wut_mk3_0 | **0.7720±0.0000** | 0.5764 | 0.5314 |
| wut_labirynth_0 | **0.7884±0.0000** | 0.5221 | 0.5221 |
| graves_parabolic_0 | **0.9802±0.0000** | 0.7068 | 0.7040 |
| sipu_d31_0 | **0.6001±0.0085** | 0.1846 | 0.1787 |
| sipu_a1_0 | **0.7499±0.0000** | 0.3269 | 0.2229 |
| sipu_r15_0 | **0.9258±0.0000** | 0.4551 | 0.2552 |
| sipu_a2_0 | **0.7555±0.0000** | 0.1685 | 0.1685 |
| sipu_a3_0 | **0.7434±0.0000** | 0.1410 | 0.1410 |
| sipu_s1_0 | **0.9888±0.0000** | 0.3695 | 0.2581 |
| sipu_s2_0 | **0.9405±0.0000** | 0.2581 | 0.2579 |

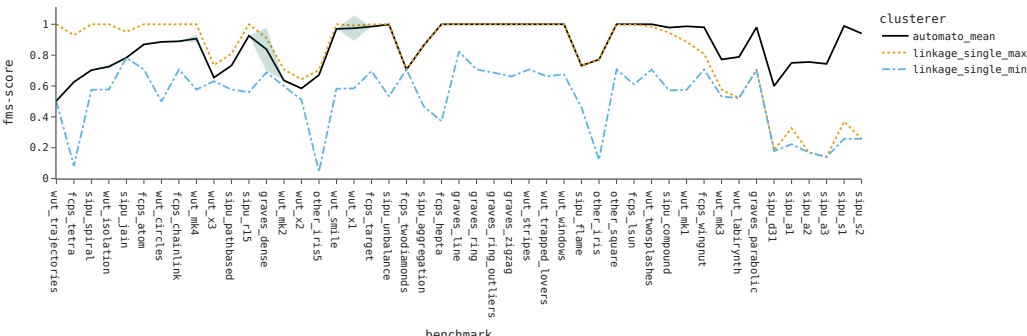

Figure 9: Comparison of AuToMATo and agglomerative clustering with single linkage.

Table 6: Fowlkes-Mallows scores of AuToMATo vs. TTK clustering algorithm

| Dataset | automato_mean | automato_std | ttk |
|---------|---------------|--------------|-----|
| wut_trajectories_0 | 0.5038±0.0107 | 0.0107 | **0.8682** |
| fcps_tetra_0 | 0.6261±0.0000 | 0.0000 | **0.9043** |
| wut_x3_0 | 0.5153±0.0000 | 0.0000 | **0.7818** |
| wut_isolation_0 | 0.7256±0.0113 | 0.0113 | **0.9416** |
| sipu_a1_0 | 0.7499±0.0000 | 0.0000 | **0.9143** |
| wut_x2_0 | 0.5846±0.0000 | 0.0000 | **0.7283** |
| sipu_flame_0 | 0.7320±0.0000 | 0.0000 | **0.8562** |
| sipu_aggregation_0 | 0.8652±0.0000 | 0.0000 | **0.9692** |
| graves_zigzag_1 | 0.6720±0.0000 | 0.0000 | **0.7698** |
| other_iris_0 | 0.7715±0.0000 | 0.0000 | **0.8639** |
| sipu_jain_0 | 0.7837±0.0000 | 0.0000 | **0.8182** |

Table 6: Fowlkes-Mallows scores of AuToMATo vs. TTK clustering algorithm

| Dataset | automato_mean | automato_std | ttk |
|---|---|---|---|
| graves_dense_0 | 0.8377±0.1396 | 0.1396 | **0.8615** |
| sipu_r15_0 | 0.9258±0.0000 | 0.0000 | **0.9374** |
| wut_mk3_0 | 0.7720±0.0000 | 0.0000 | **0.7755** |
| wut_labirynth_0 | **0.7884±0.0000** | 0.0000 | **0.7884** |
| fcps_chainlink_0 | **0.8896±0.0000** | 0.0000 | **0.8896** |
| wut_smile_0 | **0.9681±0.0000** | 0.0000 | **0.9681** |
| wut_stripes_0 | **1.0000±0.0000** | 0.0000 | **1.0000** |
| graves_ring_outliers_0 | **1.0000±0.0000** | 0.0000 | **1.0000** |
| wut_smile_1 | **0.9701±0.0000** | 0.0000 | **0.9701** |
| sipu_unbalance_0 | **0.9986±0.0008** | 0.0008 | 0.9951 |
| sipu_s1_0 | **0.9888±0.0000** | 0.0000 | 0.9843 |
| sipu_s2_0 | **0.9405±0.0000** | 0.0000 | 0.9311 |
| other_iris5_0 | **0.6712±0.0000** | 0.0000 | 0.6612 |
| fcps_atom_0 | **0.8694±0.0000** | 0.0000 | 0.8472 |
| wut_x3_1 | **0.6546±0.0000** | 0.0000 | 0.6312 |
| sipu_d31_0 | **0.6001±0.0085** | 0.0085 | 0.5667 |
| fcps_hepta_0 | **1.0000±0.0000** | 0.0000 | 0.9594 |
| graves_parabolic_1 | **0.6916±0.0000** | 0.0000 | 0.6473 |
| sipu_r15_2 | **0.4867±0.0000** | 0.0000 | 0.4322 |
| sipu_pathbased_0 | **0.6517±0.0000** | 0.0000 | 0.5947 |
| sipu_spiral_0 | **0.7028±0.0000** | 0.0000 | 0.6422 |
| sipu_r15_1 | **0.5436±0.0000** | 0.0000 | 0.4827 |
| sipu_pathbased_1 | **0.7322±0.0000** | 0.0000 | 0.6668 |
| fcps_target_0 | **0.9850±0.0000** | 0.0000 | 0.9185 |
| wut_x1_0 | **0.9741±0.0818** | 0.0818 | 0.8960 |
| wut_twosplashes_0 | **1.0000±0.0000** | 0.0000 | 0.9140 |
| wut_mk4_0 | **0.9072±0.0234** | 0.0234 | 0.8050 |
| wut_mk2_0 | **0.6356±0.0000** | 0.0000 | 0.5302 |
| graves_parabolic_0 | **0.9802±0.0000** | 0.0000 | 0.8653 |
| sipu_compound_4 | **0.9442±0.0000** | 0.0000 | 0.8145 |
| fcps_wingnut_0 | **0.9805±0.0000** | 0.0000 | 0.8497 |
| wut_circles_0 | **0.8857±0.0000** | 0.0000 | 0.7543 |
| wut_mk1_0 | **0.9866±0.0000** | 0.0000 | 0.8148 |
| fcps_twodiamonds_0 | **0.7067±0.0000** | 0.0000 | 0.5251 |
| sipu_compound_0 | **0.8616±0.0000** | 0.0000 | 0.6728 |
| sipu_compound_1 | **0.9786±0.0000** | 0.0000 | 0.7892 |
| graves_line_0 | **1.0000±0.0000** | 0.0000 | 0.7917 |
| fcps_lsun_0 | **1.0000±0.0000** | 0.0000 | 0.7897 |
| wut_trapped_lovers_0 | **1.0000±0.0000** | 0.0000 | 0.7859 |
| other_square_0 | **1.0000±0.0000** | 0.0000 | 0.7774 |
| graves_zigzag_0 | **1.0000±0.0000** | 0.0000 | 0.7686 |
| graves_ring_0 | **1.0000±0.0000** | 0.0000 | 0.7278 |
| sipu_a2_0 | **0.7555±0.0000** | 0.0000 | 0.4641 |
| wut_windows_0 | **1.0000±0.0000** | 0.0000 | 0.5853 |
| sipu_a3_0 | **0.7434±0.0000** | 0.0000 | 0.1882 |

Table 7: Fowlkes-Mallows scores of AuToMATo vs. hierarchical clustering with complete linkage

| Dataset | automato_mean | linkage_complete_max | linkage_complete_min |
|---|---|---|---|
| sipu_r15_2 | 0.4867±0.0000 | **1.0000** | 0.2256 |
| wut_trajectories_0 | 0.5038±0.0107 | **1.0000** | 0.1706 |
| sipu_r15_1 | 0.5436±0.0000 | **0.8954** | 0.2516 |
| wut_x3_1 | 0.6546±0.0000 | **0.9740** | 0.2004 |

Continued on next page

Table 7: Fowlkes-Mallows scores of AuToMATo vs. hierarchical clustering with complete linkage

| Dataset | automato_mean | linkage_complete_max | linkage_complete_min |
|---|---|---|---|
| fcps_tetra_0 | 0.6261±0.0000 | **0.9356** | 0.0651 |
| sipu_d31_0 | 0.6001±0.0085 | **0.8733** | 0.2717 |
| wut_x3_0 | 0.5153±0.0000 | **0.7842** | 0.2477 |
| sipu_a3_0 | 0.7434±0.0000 | **0.8979** | 0.2294 |
| wut_mk3_0 | 0.7720±0.0000 | **0.9207** | 0.0711 |
| wut_x2_0 | 0.5846±0.0000 | **0.7298** | 0.1964 |
| sipu_a2_0 | 0.7555±0.0000 | **0.8992** | 0.2642 |
| sipu_jain_0 | 0.7837±0.0000 | **0.9116** | 0.1288 |
| wut_circles_0 | 0.8857±0.0000 | **1.0000** | 0.1761 |
| sipu_r15_0 | 0.9258±0.0000 | **0.9799** | 0.3372 |
| sipu_a1_0 | 0.7499±0.0000 | **0.8040** | 0.3092 |
| graves_zigzag_1 | 0.6720±0.0000 | **0.7119** | 0.3039 |
| sipu_aggregation_0 | 0.8652±0.0000 | **0.9030** | 0.1246 |
| other_iris_0 | 0.7715±0.0000 | **0.8064** | 0.0680 |
| wut_x1_0 | 0.9741±0.0818 | **1.0000** | 0.2326 |
| sipu_unbalance_0 | 0.9986±0.0008 | **0.9988** | 0.5774 |
| fcps_hepta_0 | **1.0000±0.0000** | **1.0000** | 0.4321 |
| fcps_twodiamonds_0 | **0.7067±0.0000** | 0.7060 | 0.0916 |
| sipu_flame_0 | **0.7320±0.0000** | 0.7276 | 0.0834 |
| sipu_compound_0 | **0.8616±0.0000** | 0.8472 | 0.1567 |
| sipu_s1_0 | **0.9888±0.0000** | 0.9563 | 0.3672 |
| sipu_pathbased_0 | **0.6517±0.0000** | 0.6022 | 0.1384 |
| sipu_pathbased_1 | **0.7322±0.0000** | 0.6709 | 0.1539 |
| graves_parabolic_1 | **0.6916±0.0000** | 0.6168 | 0.1482 |
| sipu_compound_1 | **0.9786±0.0000** | 0.9023 | 0.1366 |
| sipu_compound_4 | **0.9442±0.0000** | 0.8652 | 0.1408 |
| graves_dense_0 | **0.8377±0.1396** | 0.7584 | 0.3538 |
| wut_mk1_0 | **0.9866±0.0000** | 0.8950 | 0.1591 |
| wut_smile_1 | **0.9701±0.0000** | 0.8697 | 0.3562 |
| graves_parabolic_0 | **0.9802±0.0000** | 0.8610 | 0.1088 |
| wut_mk2_0 | **0.6356±0.0000** | 0.5032 | 0.1096 |
| other_iris5_0 | **0.6712±0.0000** | 0.5305 | 0.0451 |
| fcps_atom_0 | **0.8694±0.0000** | 0.7278 | 0.1364 |
| wut_smile_0 | **0.9681±0.0000** | 0.8206 | 0.3793 |
| sipu_s2_0 | **0.9405±0.0000** | 0.7642 | 0.3114 |
| wut_mk4_0 | **0.9072±0.0234** | 0.7282 | 0.1297 |
| fcps_target_0 | **0.9850±0.0000** | 0.7881 | 0.1934 |
| fcps_lsun_0 | **1.0000±0.0000** | 0.7668 | 0.1377 |
| fcps_wingnut_0 | **0.9805±0.0000** | 0.7406 | 0.0816 |
| graves_ring_0 | **1.0000±0.0000** | 0.7589 | 0.1849 |
| graves_ring_outliers_0 | **1.0000±0.0000** | 0.7528 | 0.1995 |
| wut_labirynth_0 | **0.7884±0.0000** | 0.4893 | 0.1426 |
| fcps_chainlink_0 | **0.8896±0.0000** | 0.5889 | 0.0919 |
| sipu_spiral_0 | **0.7028±0.0000** | 0.3512 | 0.1424 |
| wut_isolation_0 | **0.7256±0.0113** | 0.3397 | 0.1153 |
| graves_line_0 | **1.0000±0.0000** | 0.5972 | 0.1909 |
| other_square_0 | **1.0000±0.0000** | 0.5846 | 0.1142 |
| graves_zigzag_0 | **1.0000±0.0000** | 0.5505 | 0.2042 |
| wut_twosplashes_0 | **1.0000±0.0000** | 0.5310 | 0.2771 |
| wut_stripes_0 | **1.0000±0.0000** | 0.5136 | 0.0706 |
| wut_trapped_lovers_0 | **1.0000±0.0000** | 0.4790 | 0.0579 |
| wut_windows_0 | **1.0000±0.0000** | 0.4349 | 0.0781 |

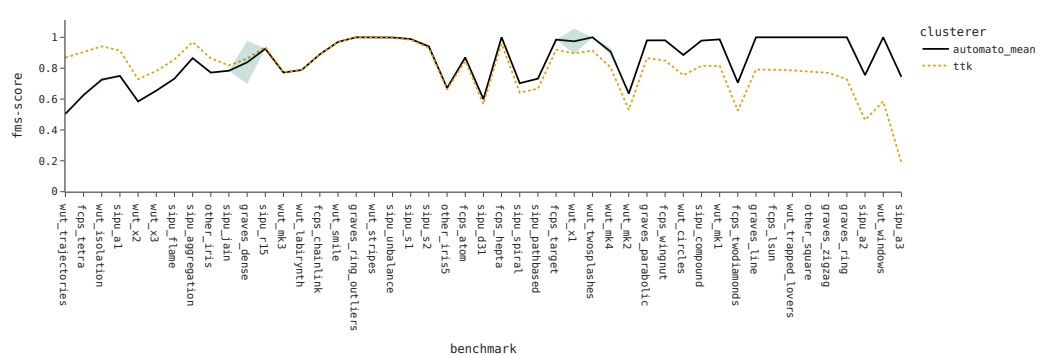

Figure 10: Comparison of AuToMATo and the TTK-algorithm.

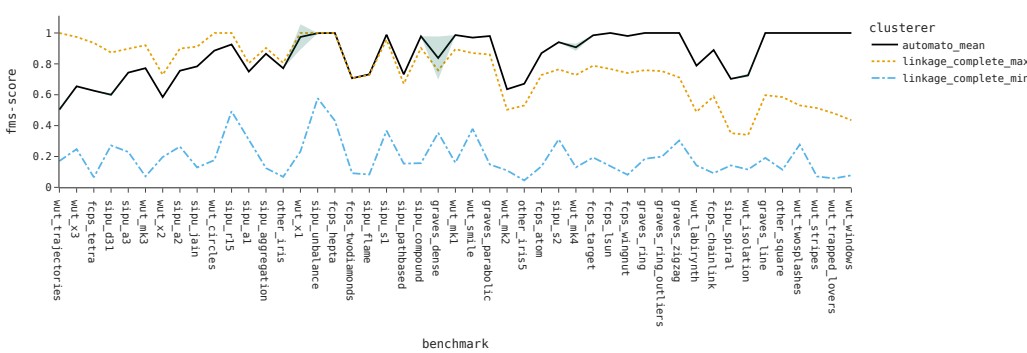

Figure 11: Comparison of AuToMATo and agglomerative clustering with complete linkage.

Table 8: Fowlkes-Mallows scores of AuToMATo vs. hierarchical clustering with Ward linkage

| Dataset | automato_mean | linkage_ward_max | linkage_ward_min |
|---|---|---|---|
| wut_x3_0 | 0.5153±0.0000 | **0.8537** | 0.2203 |
| sipu_d31_0 | 0.6001±0.0085 | **0.9223** | 0.2766 |
| sipu_a3_0 | 0.7434±0.0000 | **0.9377** | 0.2889 |
| sipu_a2_0 | 0.7555±0.0000 | **0.9360** | 0.2653 |
| sipu_a1_0 | 0.7499±0.0000 | **0.9166** | 0.2464 |
| wut_x2_0 | 0.5846±0.0000 | **0.7219** | 0.2076 |
| sipu_r15_2 | 0.4867±0.0000 | **0.5893** | 0.1868 |
| graves_zigzag_1 | 0.6720±0.0000 | **0.7358** | 0.2693 |
| sipu_r15_0 | 0.9258±0.0000 | **0.9832** | 0.4072 |
| sipu_r15_1 | 0.5436±0.0000 | **0.5993** | 0.2082 |
| wut_x3_1 | 0.6546±0.0000 | **0.7090** | 0.1795 |
| wut_x1_0 | 0.9741±0.0818 | **1.0000** | 0.2429 |
| sipu_unbalance_0 | 0.9986±0.0008 | **1.0000** | 0.2063 |
| fcps_hepta_0 | **1.0000±0.0000** | **1.0000** | 0.4314 |
| sipu_s1_0 | **0.9888±0.0000** | 0.9844 | 0.2453 |
| sipu_pathbased_0 | **0.6517±0.0000** | 0.6251 | 0.1370 |
| sipu_s2_0 | **0.9405±0.0000** | 0.9085 | 0.2177 |
| sipu_pathbased_1 | **0.7322±0.0000** | 0.6844 | 0.1523 |
| fcps_tetra_0 | **0.6261±0.0000** | 0.5622 | 0.0651 |

Table 8: Fowlkes-Mallows scores of AuToMATo vs. hierarchical clustering with Ward linkage

| Dataset | automato_mean | linkage_ward_max | linkage_ward_min |
|---|---|---|---|
| graves_dense_0 | **0.8377±0.1396** | 0.7454 | 0.2684 |
| wut_trajectories_0 | **0.5038±0.0107** | 0.3933 | 0.0981 |
| other_iris_0 | **0.7715±0.0000** | 0.6377 | 0.0680 |
| fcps_atom_0 | **0.8694±0.0000** | 0.7272 | 0.1016 |
| graves_parabolic_1 | **0.6916±0.0000** | 0.5260 | 0.1301 |
| fcps_target_0 | **0.9850±0.0000** | 0.7759 | 0.1503 |
| other_iris5_0 | **0.6712±0.0000** | 0.4508 | 0.0451 |
| sipu_aggregation_0 | **0.8652±0.0000** | 0.6064 | 0.1215 |
| sipu_flame_0 | **0.7320±0.0000** | 0.4555 | 0.0819 |
| sipu_compound_0 | **0.8616±0.0000** | 0.5846 | 0.1523 |
| wut_mk3_0 | **0.7720±0.0000** | 0.4911 | 0.0711 |
| fcps_twodiamonds_0 | **0.7067±0.0000** | 0.3967 | 0.0861 |
| sipu_jain_0 | **0.7837±0.0000** | 0.4668 | 0.1201 |
| wut_mk1_0 | **0.9866±0.0000** | 0.6564 | 0.1473 |
| fcps_lsun_0 | **1.0000±0.0000** | 0.6659 | 0.1270 |
| sipu_compound_4 | **0.9442±0.0000** | 0.5793 | 0.1368 |
| sipu_spiral_0 | **0.7028±0.0000** | 0.3100 | 0.1384 |
| wut_labirynth_0 | **0.7884±0.0000** | 0.3749 | 0.0961 |
| sipu_compound_1 | **0.9786±0.0000** | 0.5648 | 0.1327 |
| wut_twosplashes_0 | **1.0000±0.0000** | 0.5817 | 0.2046 |
| wut_smile_1 | **0.9701±0.0000** | 0.5246 | 0.2352 |
| wut_smile_0 | **0.9681±0.0000** | 0.5179 | 0.2505 |
| wut_mk2_0 | **0.6356±0.0000** | 0.1814 | 0.0997 |
| graves_zigzag_0 | **1.0000±0.0000** | 0.5448 | 0.1809 |
| wut_isolation_0 | **0.7256±0.0113** | 0.2309 | 0.0800 |
| graves_line_0 | **1.0000±0.0000** | 0.5045 | 0.1667 |
| wut_circles_0 | **0.8857±0.0000** | 0.3681 | 0.1323 |
| fcps_chainlink_0 | **0.8896±0.0000** | 0.3407 | 0.0894 |
| wut_mk4_0 | **0.9072±0.0234** | 0.3557 | 0.1079 |
| graves_parabolic_0 | **0.9802±0.0000** | 0.4184 | 0.0935 |
| graves_ring_0 | **1.0000±0.0000** | 0.4135 | 0.1427 |
| graves_ring_outliers_0 | **1.0000±0.0000** | 0.4082 | 0.1515 |
| fcps_wingnut_0 | **0.9805±0.0000** | 0.3229 | 0.0727 |
| other_square_0 | **1.0000±0.0000** | 0.3347 | 0.1007 |
| wut_windows_0 | **1.0000±0.0000** | 0.2833 | 0.0663 |
| wut_trapped_lovers_0 | **1.0000±0.0000** | 0.2552 | 0.0471 |
| wut_stripes_0 | **1.0000±0.0000** | 0.1922 | 0.0552 |

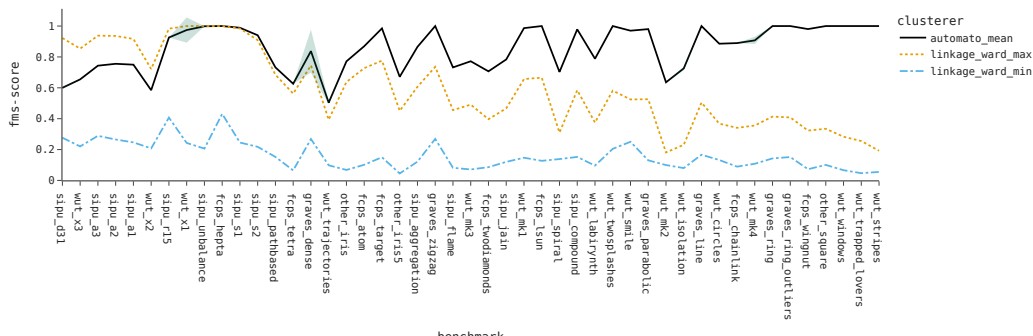

Figure 12: Comparison of AuToMATo and agglomerative clustering with Ward linkage.

Table 9: Fowlkes-Mallows scores of AuToMATo vs. FINCH

| Dataset | automato_mean | automato_std | finch |
|---|---|---|---|
| wut_x3_0 | 0.5153±0.0000 | 0.0000 | **0.7970** |
| sipu_d31_0 | 0.6001±0.0085 | 0.0085 | **0.8657** |
| sipu_a3_0 | 0.7434±0.0000 | 0.0000 | **0.8306** |
| wut_x2_0 | 0.5846±0.0000 | 0.0000 | **0.6671** |
| wut_mk3_0 | 0.7720±0.0000 | 0.0000 | **0.8503** |
| other_iris5_0 | 0.6712±0.0000 | 0.0000 | **0.7008** |
| sipu_a2_0 | 0.7555±0.0000 | 0.0000 | **0.7635** |
| wut_x3_1 | 0.6546±0.0000 | 0.0000 | **0.6619** |
| sipu_unbalance_0 | 0.9986±0.0008 | 0.0008 | **0.9998** |
| sipu_r15_0 | **0.9258±0.0000** | 0.0000 | 0.9083 |
| wut_mk1_0 | **0.9866±0.0000** | 0.0000 | 0.9655 |
| other_iris_0 | **0.7715±0.0000** | 0.0000 | 0.7477 |
| sipu_a1_0 | **0.7499±0.0000** | 0.0000 | 0.7124 |
| sipu_r15_2 | **0.4867±0.0000** | 0.0000 | 0.4156 |
| graves_zigzag_1 | **0.6720±0.0000** | 0.0000 | 0.5965 |
| graves_dense_0 | **0.8377±0.1396** | 0.1396 | 0.7615 |
| sipu_r15_1 | **0.5436±0.0000** | 0.0000 | 0.4641 |
| sipu_s1_0 | **0.9888±0.0000** | 0.0000 | 0.8728 |
| fcps_hepta_0 | **1.0000±0.0000** | 0.0000 | 0.8794 |
| fcps_atom_0 | **0.8694±0.0000** | 0.0000 | 0.7319 |
| fcps_tetra_0 | **0.6261±0.0000** | 0.0000 | 0.4680 |
| sipu_s2_0 | **0.9405±0.0000** | 0.0000 | 0.7282 |
| wut_x1_0 | **0.9741±0.0818** | 0.0818 | 0.7607 |
| graves_parabolic_1 | **0.6916±0.0000** | 0.0000 | 0.4446 |
| sipu_flame_0 | **0.7320±0.0000** | 0.0000 | 0.4767 |
| sipu_pathbased_0 | **0.6517±0.0000** | 0.0000 | 0.3440 |
| sipu_compound_0 | **0.8616±0.0000** | 0.0000 | 0.5390 |
| sipu_pathbased_1 | **0.7322±0.0000** | 0.0000 | 0.3828 |
| wut_trajectories_0 | **0.5038±0.0107** | 0.0107 | 0.1503 |
| fcps_lsun_0 | **1.0000±0.0000** | 0.0000 | 0.6206 |
| sipu_compound_4 | **0.9442±0.0000** | 0.0000 | 0.5493 |
| sipu_jain_0 | **0.7837±0.0000** | 0.0000 | 0.3824 |
| sipu_compound_1 | **0.9786±0.0000** | 0.0000 | 0.5356 |
| sipu_spiral_0 | **0.7028±0.0000** | 0.0000 | 0.2553 |
| wut_mk4_0 | **0.9072±0.0234** | 0.0234 | 0.4331 |
| wut_mk2_0 | **0.6356±0.0000** | 0.0000 | 0.1478 |
| sipu_aggregation_0 | **0.8652±0.0000** | 0.0000 | 0.3674 |
| fcps_twodiamonds_0 | **0.7067±0.0000** | 0.0000 | 0.1837 |
| wut_smile_0 | **0.9681±0.0000** | 0.0000 | 0.4452 |
| wut_smile_1 | **0.9701±0.0000** | 0.0000 | 0.4181 |
| fcps_target_0 | **0.9850±0.0000** | 0.0000 | 0.4297 |
| wut_labirynth_0 | **0.7884±0.0000** | 0.0000 | 0.2209 |
| wut_isolation_0 | **0.7256±0.0113** | 0.0113 | 0.1440 |
| wut_twosplashes_0 | **1.0000±0.0000** | 0.0000 | 0.4162 |
| graves_zigzag_0 | **1.0000±0.0000** | 0.0000 | 0.4094 |
| graves_parabolic_0 | **0.9802±0.0000** | 0.0000 | 0.3343 |
| graves_line_0 | **1.0000±0.0000** | 0.0000 | 0.3379 |
| fcps_chainlink_0 | **0.8896±0.0000** | 0.0000 | 0.2224 |
| fcps_wingnut_0 | **0.9805±0.0000** | 0.0000 | 0.3110 |
| graves_ring_0 | **1.0000±0.0000** | 0.0000 | 0.2770 |
| wut_trapped_lovers_0 | **1.0000±0.0000** | 0.0000 | 0.2767 |
| other_square_0 | **1.0000±0.0000** | 0.0000 | 0.2393 |
| graves_ring_outliers_0 | **1.0000±0.0000** | 0.0000 | 0.2248 |

Table 9: Fowlkes-Mallows scores of AuToMATo vs. FINCH

| Dataset | automato_mean | automato_std | finch |
|---|---|---|---|
| wut_circles_0 | **0.8857±0.0000** | 0.0000 | 0.0976 |
| wut_windows_0 | **1.0000±0.0000** | 0.0000 | 0.1166 |
| wut_stripes_0 | **1.0000±0.0000** | 0.0000 | 0.0867 |

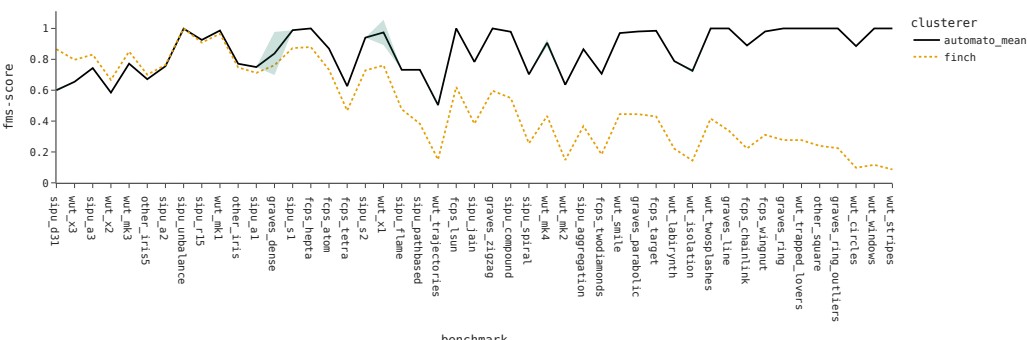

Figure 13: Comparison of AuToMATo and FINCH.

