# OpenReview forum: "AuToMATo: An Out-Of-The-Box Persistence-Based Clustering Algorithm"
_ICLR.cc/2025/Conference — ICLR 2025 Conference Withdrawn Submission_

### Official Review · Reviewer_GkFA · 2024-10-24

**Soundness:** 2
**Presentation:** 2
**Contribution:** 1
**Rating:** 3
**Confidence:** 4

**Summary:**

The paper studies the ToMATo clustering algorithm (Chazal et al. 2013) in high dimensions.
ToMATo input includes a neighbor graph (e.g. kNN graph), an estimated density for each point derived from the neighbor graph, and a threshold parameter \tau to decide merging clustering.
In principle, ToMATo first forms several clusters by connecting points with high density to its neighbor in lower density.
After that ToMATo constructs a persistence diagram which measures the size of the formed clusters (via the prominence notion reflecting number of points in the same high density region).
Using the persistent diagaram and the parameter \tau, ToMATo can provide a hierarchical clustering structure, which is similar to HDBSCAN [a].

Since the setting of \tau is sensitive, Chazal et al. 2016 proposed to use boostrap, that samples several subset X' of X to estimate the relevant value for \tau.

This works implements ToMATo algorithm with the boostrap method, and carries out experiments to demonstrate the performance of ToMATo compared to other (mainly density-based) clustering algorithms.

[a] Density-Based Clustering Based on Hierarchical Density Estimates - PKDD 2013

**Strengths:**

- The paper presents an implementation of a hierarchical clustering ToMATo.
- Some experiments and ablation study on ToMATo with Mapper, that approximates the Reeb graph of a manifold based on the sampled points, show some potentials of the implementation.

**Weaknesses:**

I feel that the novelty of the work is limited in the sense that the paper implements a clustering algorithm and presents some comparison with other clustering competitors.
Regarding the clustering accuracy, the paper use FMI scores of clustering competitors though the improvement of ToMATo is quite maginal. It would be better to use other popular measures, including AMI or NMI, since these measures are less sensitive to different number of clusters and cluster sizes.  Also, there should be the reported running time of different clustering algorithms.

The main contribution of AutoToMATo (an implementation of Tomato with boostrap) is to replace the parameter \tau by the parameter \alpha which is easier to set up in several data sets. Since the foundation of such bootstrap theory was proposed in Chazal et al. 2016, this limits the contribution of the work.

It seems that the running time complexity in Line 263 does not consider the graph construction, which require O(n^2) time in high dimensional space. Will the graph construction complexity be part of time complexity of the algorithm?

Some typos:
- Line 035: a parameters
- Line 11: must clear

**Questions:**

Q1) Could the author clarify further regarding
- The sensitity of parameter of neighbor graph (e.g k or \delta) regarding the setting of parameter \tau.
- The size and dimensionality of data sets

Q2) Are there any novel aspects or improvements in the implementation of AutoMATo that could be highlighted?

---

> ### Author Response · Authors · 2024-11-26
> **Rebuttal by Authors**
>
> The authors would like to thank the reviewer for their constructive and thorough feedback.
>
> We would first respond to the points laid out by the reviewer under weaknesses. We hope that our responses are helpful and that the reviewer will take them into consideration regarding their assessment of our article.
>
> > [...] the paper use FMI scores [...] It would be better to use other popular measures, including AMI or NMI, since these measures are less sensitive to different number of clusters and cluster sizes.
>
> We disagree with the last part of this statement. As mentioned in the paragraph starting on l. 383, we specifically chose the FM-score since that is a balanced score, while MI and e.g. the Rand indices exhibit biased behavior depending on whether the clusters in the ground truth are mostly of similar sizes or not. A reference to proofs of this can be found in our article.
>
> > [...] Since the setting of \tau is sensitive, Chazal et al. 2016 proposed to use bootstrap.
> > [...] The main contribution of AutoToMATo [...] is to replace the parameter \tau by the parameter \alpha which is easier to set up in several data sets. Since the foundation of such bootstrap theory was proposed in Chazal et al. 2016, this limits the contribution of the work.
>
> While it is true that the bootstrap on persistence diagrams was proposed by Chazal et al., those authors did not propose to use it in the context of ToMATo. Indeed, we think that one of the key contributions behind AuToMATo is that we apply the bootstrap algorithm to persistence diagrams stemming from a superlevel set filtration of an estimated density as opposed to the sublevel set filtration of some spatial function as it is done usually. In particular,
>
> > It seems that the running time complexity in Line 263 does not consider the graph construction, which require O(n^2) time in high dimensional space. Will the graph construction complexity be part of time complexity of the algorithm?
>
> Yes, we outlined this in the Appendix on l. 674 and following.
>
> In what follows, we address the points raised by the reviewer under "Questions", in the same order as they are listed in the review. We hope that our responses are helpful and that the reviewer will take them into consideration regarding their assessment of our article.
>
> **Q1:** This is a very interesting question that we think deserves to be investigated in its own right. AuToMATo is sensitive to the parameters \(k\) and \(\delta\), but quantifying this dependence is non-trivial. Indeed, it is part of ongoing research, with the goal being that a thorough understanding of this dependence will allow us to improve AuToMATo even further.
>
> **Q2:** As mentioned above already, we think that one of the key contributions behind AuToMATo is that we apply the bootstrap algorithm to persistence diagrams stemming from a superlevel set filtration of an estimated density as opposed to the sublevel set filtration of some spatial function as it is done usually. Moreover, the Mapper algorithm heavily depends on the choice of clustering algorithm, and we thus regard our initial results that illustrate the benefit of using AuToMATo in this context as valuable contributions in their own right. Finally, we regard the extensively tested, well-documented and ready-to-use code implementing AuToMATo as part of the main contribution of the paper, as it makes it easy to use AuToMATo e.g. in the context of Mapper. Indeed, we are aware of ongoing work where the Mapper algorithm with AuToMATo is being successfully applied to real-world data stemming from financial applications.

---

> > ### Comment · Reviewer_GkFA · 2024-11-26
> > **Thank for your feedback**
> >
> > After reading your rebuttal messages, I think I would not change my score. The brief reasons are below:
> >
> > - As your contribution of AutoTomato is to replace sensitive parameters (\tau) by less sensitive parameters (k or \delta), this should be a highlight subsection on the experiment section.
> >
> > - Unfortunately, I do not think an new implementation with the proposed bootstrap algorithm (by Chazal et al. 2016) is significant enough for the conference.

---

### Official Review · Reviewer_SetF · 2024-10-31

**Soundness:** 2
**Presentation:** 2
**Contribution:** 1
**Rating:** 3
**Confidence:** 5

**Summary:**

The paper introduces a nice clustering algorithm based on density and persistence. It uses similar concepts as HDBSCAN and Density peaks and works with a set of predefined default parameters.

**Strengths:**

S1) Easy to follow, written clearly.
S2) Good idea and important concepts that are used
S3) Good reasoning and background information behind the choices (for experiments as well as design-choices in the algorithm development)

**Weaknesses:**

W1) Experiments are not sufficient.
a) Even though Fowlkes Mallows is a good evaluation measure, state-of-the-art papers for clustering usually include the NMI and ARI values, so please include at least one of them additionally in the appendix.
b) Presentation of the experiments is hard to follow: In the main paper, only the average over all datasets is given, which is not enough to get an idea where the algorithm is good and where not. Instead of comparing AuToMATo with competitors individually, please give an overview of all methods, but for the individual datasets.
c) Please provide an overview of the properties of the datasets. Which dimensionality, which size?
d) Regarding Figure 2 : The worst clustering result that DBSCAN returns is completely irrelevant, leave it away. You can always find an epsilon or minpts such that all points are clustered together into one cluster.
e) Include an evaluation of noise/outlier labels given by your method or exclude it from the paper entirely.
f) Do not set a seed for the experiments. That hinders evaluation of robustness.
g) In line 440 you write DBSCAN "sometimes" outperforms AuToMATo, even though it does so for around a third of the datasets. Plus, the range of epsilon values is not chosen according to best practices (see, e.g., Schubert, E., Sander, J., Ester, M., Kriegel, H. P., & Xu, X. (2017). DBSCAN revisited, revisited: why and how you should (still) use DBSCAN. ACM Transactions on Database Systems (TODS), 42(3), 1-21.)

W2) Lack of novelty and discussion. Even though the method has some promising ideas, it is very similar to HDBSCAN and Density Peaks without discussing the differences enough or showing where exactly they make a difference in practice. Synthetic datasets could help to show these differences. Furthermore, AuToMATo does not significantly outperform any of the competitors. Please elaborate for which type of datasets one should use AuToMATo over existing methods.

W3) The related work section is missing. Please add additionally to the background about ToMATo in Section 2 also background about other related methods. Elaborate on similarities, e.g., your concept of persistence is very similar to the stability used in HDBSCAN and the hill climbing approach is similar to density-peaks.

W4) Presentation: Increase font size for Fig.2 . Instead of listing all the Tables in the appendix, make a visual overview where readers can compare all your competitors for the different datasets.

Typos: line 060, 354/355, 420,

**Questions:**

See weak points.

Q1) In which range are the dimensionalities of your tested datasets?

---

### Official Review · Reviewer_zgxT · 2024-11-01

**Soundness:** 2
**Presentation:** 2
**Contribution:** 3
**Rating:** 5
**Confidence:** 3

**Summary:**

This paper introduces AuToMATo, a persistence-based clustering algorithm based on the existing topological clustering approach, ToMATo. The methodology involves executing a bottleneck bootstrap on the persistence diagram generated by ToMATo to distinguish significant from non-significant peaks of the estimated density function.
Although AuToMATo is not entirely parameter-free, the authors propose default settings with great performance without tuning the parameters. For evaluation, the algorithm was benchmarked using data sets from the Clustering Benchmarks suite, comparing it with DBSCAN, HDBSCAN, hierarchical clustering (employing Ward, single, complete, and average linkage strategies), the FINCH clustering algorithm, and a TTK-based algorithm stemming from the Topology ToolKit suite. Experiments showed that AuToMATo performs better than parameter-free clustering algorithms and other algorithms. The authors also presented an application of AuToMATo in combination with Mapper using synthetic two-dimensional data and the Miller-Reaven diabetes dataset.

**Strengths:**

The paper presents AuToMATo, a new and improved version of the topological clustering algorithm ToMATo.

The authors provide a rigorous explanation of the algorithm, including detailed mathematical definitions.

The paper is generally well-written.

AuToMATo achieves competitive performance without the need for manual parameter tuning.

**Weaknesses:**

The experiments only use datasets from the Clustering Benchmarks suite. Including more high-dimensional and real-world datasets would better evaluate the AuToMATo algorithm's scalability and performance.

The paper does not provide enough experiments and discussion comparing AuToMATo to other parameter-free clustering algorithms, which is necessary to demonstrate its effectiveness for this contribution.

->The paper does not sufficiently examine how changes in parameters affect the algorithm. While it's mentioned that the choice of alpha and B is justified by experiments, there should be a discussion that provides more evidence and reasoning. Including additional experiments that demonstrate how different values of alpha and B influence the clustering results would improve the discussion.

->Relying only on the Fowlkes-Mallows score is not sufficient. Including other evaluation metrics (internal and external) would provide a more complete evaluation and help compare with other algorithms.

-> The application of AuToMATo to Mapper is briefly mentioned but needs a more detailed explanation of the methods and the results obtained.

**Questions:**

page 5 -> Anonymized GitHub-link but no link. Where is the code?

Relying only on the Fowlkes-Mallows score is not sufficient. Including other evaluation metrics (internal as DB and external as NMI...) would provide a more complete evaluation and help compare with other algorithms.

In the context of the proposed approach, there have been no comparisons with non-parametric algorithms. Given the search for topological structure, I think it would be interesting to compare with topological algorithms such as SOM and spectral clustering.

---

> ### Author Response · Authors · 2024-11-26
> **Rebuttal by Authors**
>
> The authors would like to thank the reviewer for their constructive and thorough feedback.
>
> In what follows, we address some of the points raised by the reviewer under "Weaknesses" and "Questions", in the same order as they are listed in the review. We hope that our responses are helpful and that the reviewer will take them into consideration regarding their assessment of our article.
>
> > The application of AuToMATo to Mapper is briefly mentioned but needs a more detailed explanation of the methods and the results obtained.
>
> We agree that the section on the application of AuToMATo to Mapper is a bit terse; we had to make this choice due to space constraints. Furthermore, a thorough quantitative analysis of Mapper+AuToMATo is part of ongoing research and we thus formulated the section on the application of AuToMATo to Mapper as a outlook showcasing first promising results. Indeed, we are aware of ongoing work where the Mapper algorithm with AuToMATo is being successfully applied to real-world data stemming from financial applications.
>
> **Q1:** The link is anonymized since the author's GitHub contains identifying information. We therefore included the code in the Supplementary Material of our submission as a ready-to-use Python package.
>
> **Q2:** As indicated in the article (l. 386), we did not include NMI or any Rand scores since those exhibit biased behavior depending on whether the clusters in the ground truth are mostly of similar sizes or not. Moreover, we did not include the DB score since the value of that is undefined if the clustering found consists of a single cluster, and in our extensive analysis of the various algorithms it does happen that e.g. certain parameter selection for DBSCAN result in a single cluster. We do acknowledge, however, that we failed to include this reasoning in the submitted paper; we would be happy to include this in an updated version.
>
> **Q3:** While we agree that including more clustering algorithms would be interesting, we do not understand the reviewer's claim that "there have been no comparisons with non-parametric algorithms". We do compare AuToMATo with e.g. FINCH and the TTK algorithm, both of which are non-parametric and, moreover, the latter of which builds on ideas from topology. Moreover, we are puzzled by the statement that spectral clustering is a topological algorithm; it relies on ideas involving the spectrum of matrices and, in particular, we chose not to include it in our analysis since, unlike AuToMATo and the reference algorithms, it requires the user to specify the number of clusters in advance, which defies the purpose of comparing out-of-the-box algorithms against each other.

---

### Official Review · Reviewer_4i4K · 2024-11-01

**Soundness:** 3
**Presentation:** 3
**Contribution:** 2
**Rating:** 6
**Confidence:** 3

**Summary:**

This paper considers and extends the ToMATo clustering algorithm, which finds clusters based on the persistent homology of a density estimate of the data. This paper proposes a novel method to automate the selection of the $\tau$ parameter of the ToMATo algorithm, giving a parameter-free, out-of-the-box clustering algorithm.

The new algorithm is based on the 'bottleneck bootstrap' process in order to build a persistence diagram in which the prominent connected components can be easily identified.

**Strengths:**

This paper is extremely clear, and presents a novel algorithm which successfully solves the problem of parameter selection in the ToMATo algorithm. The newly presented algorithm is easy to use and likely to be an effective drop-in replacement for the ToMATo algorithm.

**Weaknesses:**

The precise novel contribution of this paper is not completely clear. Given that the bottleneck bootstrap process was defined by Chazal et al. (2017), what is the key new insight that enables the AuToMATo algorithm to work? This should be made more clear in the write-up.

The datasets used for the experimental evaluation seem to be generally quite small (< 10000 data points), low dimensional synthetic datasets. It would be interesting to see a comparison on some larger, real world datasets. For example, the mnist dataset (which is one of the datasets in the benchmark used) seems to have been excluded.

Additionally, the running time of the algorithms are not reported in the experimental section. While I understand that there is often a trade-off between running time and performance, it would be interesting to see this trade-off explicitly discussed.

Finally, although the choice to use the Fowkes-Mellows score in the evaluation is justified in the paper, given that the Adjusted Rand Index and Normalised Mutual Information are very standard in the literature, in my view it would be better to compare on all metrics.

**Questions:**

* What is the key novel insight behind the AuToMATo algorithm?
* How does the empirical running time of AuToMATo compare with the alternative algorithms?
* Is there hope to scale AuToMATo to large, high-dimensional real-world datasets?

---

> ### Author Response · Authors · 2024-11-26
> **Rebuttal by Authors**
>
> The authors would like to thank the reviewer for their constructive, thorough and positive feedback.
>
> In what follows, we address the points raised by the reviewer under "Questions", in the same order as they are listed in the review. We hope that our responses are helpful and that the reviewer will take them into consideration regarding their assessment of our article.
>
> **Q1:** While the bootstrapping procedure is indeed an existing algorithm, the key insight behind AuToMATo is that we apply that algorithm to persistence diagrams stemming from a superlevel set filtration of an estimated density as opposed to the sublevel set filtration of some spatial function as it is the case usually. Moreover, we regard the extensively tested, well-documented and ready-to-use code implementing AuToMATo as part of the main contribution of the paper, as it makes it easy to use AuToMATo e.g. in the context of Mapper. Indeed, we are aware of ongoing work where the Mapper algorithm with AuToMATo is yields promising results when applied to real-world data stemming from financial applications.
>
> **Q2:** The runtime of AuToMATo is usually larger than that of the other clustering algorithms. This is due to the fact that the bootstrapping procedure is quite expensive. On the other hand, this aspect of the algorithm can by parallelized to a theoretically arbitrary extent which, depending on the hardware, decreases the runtime dramatically.
>
> **Q3:** Yes. As mentioned above, we know of ongoing work where the Mapper algorithm with AuToMATo is being successfully applied to real-world data stemming from financial applications.

---

> > ### Comment · Reviewer_4i4K · 2024-11-26
> > **Thanks**
> >
> > Thanks for your response. I have read it and will keep my (positive) score. Given that the implementation comprises a large part of the contribution of this paper, in my view providing a more extensive experimental section, including real-world and high-dimensional datasets, is the main way in which the paper could be improved.

---

### Note · Authors · 2025-01-23

I have read and agree with the venue's withdrawal policy on behalf of myself and my co-authors.